# Systematic review and meta-analysis of the effectiveness of pre-pregnancy care for women with diabetes for improving maternal and perinatal outcomes

Hayfaa A. Wahabi[1,2], Amel Fayed[3,4]*, Samia Esmaeil[1], Hala Elmorshedy[3,4], Maher A. Titi[1,5], Yasser S. Amer[1,6], Rasmieh A. Alzeidan[7], Abdulaziz A. Alodhayani[2], Elshazaly Saeed[8], Khawater H. Bahkali[9], Melissa K. Kahili-Heede[10], Amr Jamal[1,2], Yasser Sabr[11]

1 Research Chair for Evidence-Based Healthcare and Knowledge Translation, King Saud University, Riyadh, Saudi Arabia, 2 Department of Family and Community Medicine, King Saud University Medical City and College of Medicine, Riyadh, Saudi Arabia, 3 College of Medicine, Clinical Department, Princess Nourah Bint Abdulrahman University, Riyadh, Saudi Arabia, 4 High Institute of Public Health, Alexandria University, Alexandria, Egypt, 5 Patient Safety Unit, Quality Management Department, King Saud University Medical City, Riyadh, Saudi Arabia, 6 Clinical Practice Guidelines Unit, Quality Management Department, King Saud University Medical City, Riyadh, Saudi Arabia, 7 Cardiac Science Department, College of Medicine, King Saud University, Riyadh, Saudi Arabia, 8 Prince Abdulla bin Khaled Coeliac Disease Research Chair, King Saud University, Riyadh, Saudi Arabia, 9 Saudi Center for Disease Prevention and Control, Riyadh, Saudi Arabia, 10 John A. Burns School of Medicine, Health Sciences Library, University of Hawaii at Manoa, Honolulu, HI, United States of America, 11 Department of Obstetrics and Gynecology, College of Medicine, King Saud University, Riyadh, Saudi Arabia

* Fayedam_200@hotmail.com

**Data Availability Statement:** All relevant data are within the manuscript and its Supporting Information files.

## Abstract

### Background

Pre-gestational diabetes mellitus is associated with increased risk of maternal and perinatal adverse outcomes. This systematic review was conducted to evaluate the effectiveness and safety of pre-conception care (PCC) in improving maternal and perinatal outcomes.

### Methods

Databases from MEDLINE, EMBASE, WEB OF SCIENCE, and Cochrane Library were searched, including the CENTRAL register of controlled trials, and CINHAL up until March 2019, without any language restrictions, for any pre-pregnancy care aiming at health promotion, glycemic control, and screening and treatment of diabetes complications in women with type I or type II pre-gestational diabetes. Trials and observational studies were included in the review. Newcastle-Ottawa scale and the Cochrane collaboration methodology for data synthesis and analysis were used, along with the GRADE tool to evaluate the body of evidence.

### Results

The search identified 8500 potentially relevant citations of which 40 reports of 36 studies were included. The meta-analysis results show that PCC reduced congenital malformations

**Funding:** This study was funded by the Deanship of Scientific Research, at Princess Nourah Bint Abdulrahman University through the fast track programme. The funder had no role in study design, data collection, data analysis, decision to publish or preparation of the manuscript.

**Competing interests:** The authors have declared that no competing interests exist.

risk by 71%, (Risk ratio (RR) 0.29; 95% CI: 0.21–0.40, 25 studies; 5903 women; high-certainty evidence). The results also show that PCC may lower HbA1c in the first trimester of pregnancy by an average of 1.27% (Mean difference (MD) 1.27; 95% CI: 1.33–1.22; 4927 women; 24 studies, moderate-certainty evidence). Furthermore, the results suggest that PCC may lead to a slight reduction in the risk of preterm delivery of 15%, (RR 0.85; 95% CI: 0.73–0.99; nine studies, 2414 women; moderate-certainty evidence). Moreover, PCC may result in risk reduction of perinatal mortality by 54%, (RR 0.46; 95% CI: 0.30–0.73; ten studies; 3071 women; moderate-certainty evidence). There is uncertainty about the effects of PCC on the early booking for antenatal care (MD 1.31; 95% CI: 1.40–1.23; five studies, 1081 women; very low-certainty evidence) and maternal hypoglycemia in the first trimester, (RR 1.38; 95% CI: 1.07–1.79; three studies; 686 women; very low- certainty evidence). In addition, results of the meta-analysis indicate that PCC may lead to 48% reduction in the risk of small for gestational age (SGA) (RR 0.52; 95% CI: 0.37–0.75; six studies, 2261 women; moderate-certainty evidence). PCC may reduce the risk of neonatal admission to intensive care unit (NICU) by 25% (RR 0.75; 95% CI: 0.67–0.84; four studies; 1322 women; moderate-certainty evidence). However, PCC may have little or no effect in reducing the cesarean section rate (RR 1.02; 95% CI: 0.96–1.07; 14 studies; 3641 women; low-certainty evidence); miscarriage rate (RR 0.86; 95% CI: 0.70–1.06; 11 studies; 2698 women; low-certainty evidence); macrosomia rate (RR 1.06; 95% CI: 0.97–1.15; nine studies; 2787 women, low-certainty evidence); neonatal hypoglycemia (RR 0.93; 95% CI: 0.74–1.18; five studies; 880 women; low-certainty evidence); respiratory distress syndrome (RR 0.78; 95% CI: 0.47–1.29; four studies; 466 women; very low-certainty evidence); or shoulder dystocia (RR 0.28; 95% CI: 0.07–1.12; 2 studies; 530 women; very low-certainty evidence).

## Conclusion

PCC for women with pre-gestational type 1 or type 2 diabetes mellitus is effective in improving rates of congenital malformations. In addition, it may improve the risk of preterm delivery and admission to NICU. PCC probably reduces maternal HbA1C in the first trimester of pregnancy, perinatal mortality and SGA. There is uncertainty regarding the effects of PCC on early booking for antenatal care or maternal hypoglycemia during the first trimester of pregnancy. PCC has little or no effect on other maternal and perinatal outcomes.

## Introduction

Globally, the burden of diabetes is increasing. The number of adults living with diabetes is expected to increase from 429 million to 629 million by the year 2045—which is almost a 50% increase in the number of the affected population [1]. Furthermore, in low and middle-income countries, the burden of diabetes is higher among the younger population, including women in the reproductive age group [2]. If the current situation remains unabated, a substantial increase in high risk pregnancies complicated with pregestational diabetes will create major health care problems in low income countries due to the higher mortality and morbidity associated with pregestational diabetes compared to non-diabetic pregnancies.

Hyperglycemia in early pregnancy increases the risk of congenital abnormalities by nine-fold compared to the normoglycemic population [3]. There is a fivefold increase in the rate of

cardiovascular abnormalities and a twofold increase in the rate of neural tube and urinary tract defects in infants of mothers with diabetes compared to the background population [4, 5]. Congenital defects and preterm births [6] were the main contributors to the high rate of perinatal mortality observed in pregnancies complicated by maternal pregestational diabetes [7, 8].

Many of the serious complications of pregestational diabetes can be averted by implementing preconception care (PCC) [9]. Education about the interaction between diabetes and pregnancy, family planning combined with diabetes self-management skills can achieve optimum glycemic control during early pregnancy, which can reduce rates of congenital abnormalities and perinatal mortality [10].

Other essential elements of PCC include; folic acid supplementation [11], lifestyle modification (weight reduction, smoking cessation), multidisciplinary medical care (endocrinologist, obstetrician, dietitian and midwives specialized in diabetes), and substituting teratogenic medications for safer ones [12].

Despite the proven clinical value and cost-effectiveness of PCC [13], there is low uptake of the service in some communities and lack of it in others. Most pregnancies are unplanned, which makes PCC unfeasible for almost 40% of women with pregestational diabetes [14]. In addition, the deprived socioeconomic status in low income countries plays a part in access and utilization of PCC [15], which puts a considerable proportion of women with diabetes at risk of adverse pregnancy outcomes.

Since the publication of our last systematic review on the effectiveness of PCC in improving maternal and perinatal outcomes, many studies have been published to investigate different interventions and outcomes of PCC [9, 16]. Additionally, with the increased recognition of the importance of evaluation of the body of evidence a grading tool, Grading of Recommendations Assessment, Development and Evaluation (GRADE), has been introduced to facilitate evidence-based decision making for interventions in clinical medicine and health policy [17].

The objectives of this systematic review are to assess the effectiveness of PCC comprehensively in improving maternal and perinatal outcomes and to evaluate the grade of the body of evidence for each outcome.

## Methods

### Search methods

A structured literature search was undertaken to review all the literature published up to March 2019. The search strategy was developed with the help of library and information retrieval specialist. We searched the following databases: MEDLINE, EMBASE, WEB OF SCIENCE, CINHAL and Google Scholar; (For full search strategy, see **S1 File**). Additionally, bibliographies of retrieved articles were manually searched for potentially relevant papers. No language or date restrictions were applied in the search.

### Study selection

The following criteria were applied for eligibility:

- Randomized and quasi-randomized controlled trials, cluster-randomized trials, and observational (cohort, cross-sectional and case control) studies were eligible for inclusion.

- Studies and trials which compared the frequency of maternal and perinatal adverse outcomes in women with diabetes who received PCC with those who did not receive PCC.

- Women of reproductive age with pregestational diabetes type 1 or type 2 diabetes mellitus who were not pregnant at the time of intervention.
- PCC interventions including (i.e. either as sole intervention or in combination):
  - Glycemic control by insulin and/or diet and/or oral hypoglycemic drugs.
  - Women counselling and/or education about diabetes complications during pregnancy, the importance of glycemic control and self-monitoring of blood glucose level.
  - Preconception screening and treatment of complications of diabetes
  - The use of contraception until optimization of glycemic control is achieved
  - Intake of multivitamin or folic acid in the preconception period.
  - Physical exercise and/or weight control.
- Studies reporting maternal and neonatal outcomes as follows:

  Maternal outcomes:
- Hemoglobin A1c (HbA1c) level in the first trimester of pregnancy
- Gestation age (GA) at the time of the first visit to antenatal care clinic (booking visit)
- Miscarriage or termination of pregnancy due to congenital abnormalities
- Induction of labor due to maternal complications of diabetes
- Delivery by cesarean section (CS) or instrumental delivery
- Maternal hypoglycemia in the first trimester

  Neonatal outcomes:
- Preterm delivery
- Congenital malformations related to maternal diabetes
- Perinatal mortality (stillbirth and neonatal death)
- Birth trauma
- Admission to neonatal intensive care unit (NICU)
- Respiratory distress syndrome (RDS)
- Macrosomia (birth weight $\geq$ 4 kg for term infants or large for gestational age (LGA)birth weight $\geq$ 90th percentile for the gestation age)
- Small for gestational age (SGA) (birth weight below the 10th percentile for the gestational age)
- Shoulder dystocia
- Neonatal hypoglycemia

## Study identification

We screened titles and abstracts of all the potential studies identified as a result of the search by two reviewers independently. Disagreements were resolved through discussion or after consultation with a third reviewer when needed.

Articles with the criteria below were excluded from the review:

- Did not contain a complete description of the study or study population

- Did not report original data (commentary, review or editorial) or reports of conference proceedings or abstracts when complete data could not be retrieved from the authors

- Participants were not women with pregestational diabetes or were pregnant at the time of intervention

- Did not assess impact of a PCC intervention

- Did not include comparatives arms.

Then the full-text papers were retrieved, and potentially relevant studies were assessed independently by two authors for eligibility by application of the inclusion/exclusion criteria. The review was registered in PROSPERO (registration number CRD42019114336) [18].

## Data extraction

The data were subsequently extracted from included studies by two reviewers using a purposefully designed data extraction form. The reviewers were not masked to the articles' authors, journals, or institutions. The data extracted were: country and year of publication, settings, study design, study duration, study population details of intervention/s and control, and outcomes. When information regarding any of the above was unclear, the authors were contacted to provide the missing details. Any disagreement on value or type of data extracted between reviewers was resolved through discussion or by consulting a third reviewer.

## Quality assessment

**Assessment of risk of bias.** Two reviewers independently assessed the risk of bias for each cohort/ case control study using The Newcastle-Ottawa Scale (NOS) [19]. The criteria assessed for cohort studies were: participants' selection, comparability of groups and assessment of outcome. While participants' selection, comparability of groups, and exposure criteria were used to assess the case-control studies. The maximum number of stars awarded for any study were nine: four stars awarded for selection of participants (exposed and non- exposed), ascertainment of exposure and temporal relation between exposure and outcome, two stars were awarded for comparability, if analysis controlled for confounding factors, and three stars were awarded for outcomes if the length of follow up was adequate, with no attrition bias, and if the outcomes were assessed independent of exposure. Studies at "high risk of bias" score less than six stars or scores no stars in comparability domain irrespective of the number of stars scored. Any difference in grading of studies was reconciled by discussion or by involving a third reviewer. For randomized controlled trials, we used the Cochrane tool for bias assessment [20].

**Overall risk of bias for outcomes.** We made explicit judgements about whether studies included in the meta-analysis of each of the main outcomes, were at high risk of bias according to the NOS criteria. We assessed the likely magnitude and direction of the bias and whether the likelihood of having an impact on the findings was of any significance.

**Publication bias.** We assessed the presence of publication bias using Funnel Plots of effect size against standard error for each meta-analysis that included ten or more studies according to Cochrane collaboration methodology. Three analyses were eligible for publication bias assessment including: the effect of PCC on congenital malformations (25 studies), HbA1C (24 studies) and perinatal mortality (ten studies). The vertical axis of the plot represents the

standard error, while the horizontal axis represents the logarithmic scale of risk ratio for dichotomous variables in case of congenital malformations and perinatal mortality. In the case of continuous variables, as in HbA1c, the horizontal axis represents the standardized mean difference. Furthermore, we assessed selective reporting in all outcomes [20].

**Assessment of the quality of the evidence.** The overall quality and strength of evidence for the main outcomes were assessed using the GRADE approach [21]. We created a 'Summary of findings' tables for the main outcomes of the review. The body of evidence is downgraded from 'high quality' by one level for serious (or by two levels for very serious) limitations depending on assessments of risk of bias, indirectness of evidence, inconsistency, imprecision of effect estimates or potential publication bias. Subsequently, the quality of evidence was graded as 'high', 'moderate', 'low' or 'very low' certainty. Evidence derived from observational studies receive an initial grade of 'low', however, we upgraded the quality of evidence when there was a large magnitude of effect (RR>2 or RR<0.5, in the absence of plausible confounders) [21]. We downgraded scores for risk of bias (weight of studies show risk of bias as assessed by low NOS <6), inconsistency (unexplained heterogeneity), indirectness of evidence (presence of factors that limit the generalizability of the results), imprecision in the pooled risk estimate (the 95% CI for risk estimates are wide or cross a minimally important difference of 10% for benefit or harm (RR 0.9–1.1)), and publication bias (evidence of small-study effects) [21]. We used the GRADEpro tool in order to create the 'Summary of findings' tables [22]. We assessed the quality of the body of evidence relating to the following outcomes for the main comparison, PCC versus no PCC; 1) Congenital malformations 2) HbA1c in the first trimester of pregnancy 3) Perinatal mortality 4) Preterm delivery 5) Maternal hypoglycemia 6) Gestational age at booking for antenatal care.

We produced a summary of the intervention effect using the GRADE approach, a measure of quality for each of the above outcomes.

**Data synthesis.** A statistical analysis using RevMan 5 software (RevMan 2014) was carried out [23]. We used the fixed-effect model to conduct meta-analyses. The pooled statistics was reported as either relative risk (RR) for categorical variables, or mean difference (MD) for continuous variables in the comparison between the intervention and control groups with 95% confidence intervals (CI). Heterogeneity was quantified in each meta-analysis using the $Tau^2$, $I^2$ and $Chi^2$ statistics [24]. We regarded heterogeneity as substantial if $I^2$ was $\geq$ 50% and either $Tau^2$ was greater than zero, or there was a low $p$ value (less than 0.10) in the $Chi^2$ test for heterogeneity.

We conducted sensitivity analyses by excluding studies with high risk of bias from the met-analysis for the main review outcomes. We conducted sensitivity analysis for two maternal outcomes which are gestation age at the first antenatal visit and first trimester HBA1c level, in addition to five neonatal outcomes, including: congenital malformations, preterm delivery, perinatal mortality, SGA, and admission to NICU.

**Differences between the protocol and the review.** The authors decided to utilize the modified version of NOS proposed by the Agency for Healthcare Research and Quality (AHRQ), U.S. Department of Health and Human Services [25, 26] as it was more rigorous, robust, and user-friendly than the original version proposed by the University of Ottawa [19].

We conducted additional sensitivity analysis by excluding studies with high risk of bias from certain outcomes

## Results

### Literature search

Our initial search identified 8500 potentially relevant citations of which 76 full text articles were reviewed (Fig 1). We identified 40 reports of 36 studies for inclusion in the analysis.

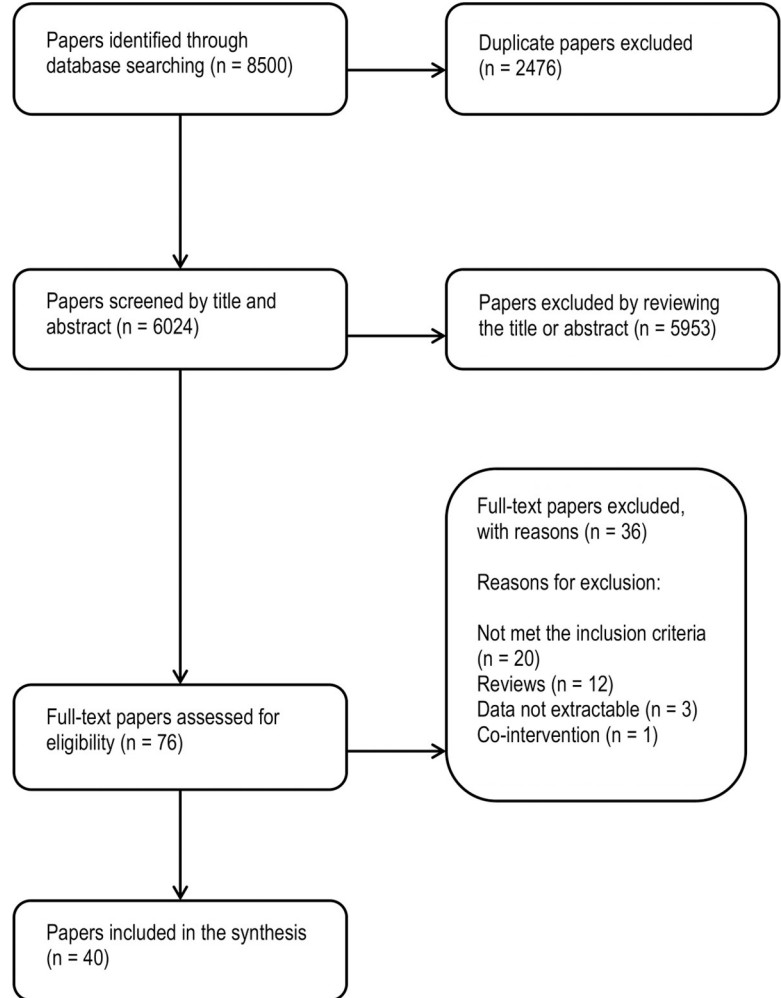

**Fig 1. Process of selection of the studies for the systematic review (PRISMA flow chart).**

Among these reports, three articles described the same cohort study with two interim [27, 28] and one final report [29], one study reported the outcomes of the same cohort in two articles [30, 31] and two articles reported the outcomes of one cohort with one interim [32] and one final report [33].

Thirty-six studies were excluded because: they did not meet the inclusion criteria, or were review articles, data were not extractable in three studies, and one report was excluded because of co-intervention applied at national level [34].

## Study characteristics

**Participants.** The main characteristics of included studies are summarized in Tables 1–3. There were 36 included studies conducted through March 2019, all of which were conducted in high income countries [35]. Of the included studies 18 were prospective cohort studies [31, 36–52], 16 were retrospective cohort studies [29, 32, 33, 49, 53–65], one was a trial [66], and one was a case control study [67]. The number of participants of cohort studies were 8199 women, among whom 3213 received PCC followed by antenatal care, whilst 4986 only received antenatal care. There were 24 participants in the case control study [67] and 180

**Table 1. Characteristics of included cohort studies.**

| Serial No. | Study ID | Participants | Intervention | Outcomes |
|---|---|---|---|---|
| 1 | **Boulot** (2003) France [45] | **PCC:** **175** women with (DM-I and DM-II) **NO-PCC:** **260** women with (DM-I and DM-II) | PCC included: • Educational delivered by health care of professionals, • Assessment of diabetes complications, • Advice regarding blood glucose optimization, • Dietary modification, • Self-monitoring of blood glucose levels, and insulin therapy. | **PCC:** **Perinatal mortality:** (3/175) **Congenital malformations:** (2/175) **NO-PCC:** **Perinatal mortality:** (16/260) **Congenital malformations:** (16/260) |
| 2 | **Cousins** (1991) USA[37] | **PCC:** **27** women with (DM-I and DM-II) **NO-PCC:** **347** women with (DM-I and DM-II) received care after conception | PCC included: • A multidisciplinary team approach to care (physicians, Diabetes educators, dietitians and social workers), • Comprehensive education, • Active self- management (e.g. self-glucose monitoring, home testing for ketone- urea, insulin injection techniques), • Routine maternal care elements and laboratory tests, • History and physical examination. | **PCC:** **Congenital malformations:** (0/27) **NO-PCC:** **Congenital malformations:** (23/347) |
| 3 | **Damm** (1989) Denmark [36] | **PCC:** **197** women with (DM-I) **NO-PCC:** **61** women with (DM-I) | PCC included: • Optimization of diabetic control at the time of conception and nidation and during the first trimester, • Pregnancy planning and contraceptive guidance. | **PCC:** **First trimester HbA1c:** 7.1 ± SD 1.2 (N = 64) **Congenital malformations:** (2/197) **NO-PCC:** **First trimester HbA1c:** 7.3 ± SD 1.5 (N = 21) **Congenital malformations:** (5/61) |
| 4 | **Dicker** (1988) Israel[55] | **PCC:** **59** women with (DM-I) **NO-PCC:** **35** women with (DM-I) | PCC included: • Insulin and dietary glycemic control, • Advice on contraception, • Screening for diabetes complications. | **PCC:** **x First trimester HbA1c:** 7.39 ± SD 0.33 (N = 59) **Miscarriage:** (5/59) **NO-PCC:** **x First trimester HbA1c:** 10.49 ± SD 0.48 (N = 35) **Miscarriage:** (10/35) |
| 5 | **Egan** (2016) Ireland [50] | **PCC:** **149** women with (DM-I and DM-II) **NO-PCC:** **265** women with (DM-I and DM-II) | PCC included: • Patient education, • Full medication review, • Assessment and treatment of diabetes complications and thyroid status. • Folic acid supplement, • Intensive glucose monitoring with a target HbA1c of less than 6.1%, • Dietary advice and Pregnancy planning. | **PCC:** **First trimester HbA1c:** 6.8 ± SD 1.2 (N = 149) **CS delivery:** (85/149) **Congenital malformations:** (1/149) **Miscarriage:** (25/149) **Instrumental delivery:** (11/149) **Maternal hypertension:** (25/149) **Preeclampsia:** (13/149) **Preterm delivery:** (17/149) **Serious adverse outcome:** (3/149) **Shoulder dystocia:** (0/149) **# Perinatal mortality:** (2/149) **¯LGA:** (49/149) **SGA:** (5/149) **Excessive GWG:** (61/149) **Neonatal hypoglycemia:** (15/149) **NICU admission:** (54/149) **NO-PCC:** **First trimester HbA1c:** 7.7 ± SD 1.8 (N = 265) **CS delivery:** (142/265) **Congenital malformations:** (12/265) **Miscarriage:** (36/265) **Instrumental delivery:** (26/265) **Maternal hypertension:** (54/265) **Preeclampsia:** (28/265) **Preterm delivery:** (47/265) **Serious adverse outcome:** (24/265) **Shoulder dystocia:** (6/265) **# Perinatal mortality:** (8/265) **¯LGA:** (75/265) **SGA:** (20/265) **Excessive GWG:** (80/265) **Neonatal hypoglycemia:** (22/265) **NICU admission:** (137/265) |
| 6 | **Cyganek** (2010) Poland [58] | **PCC:** **116** women with (DM-I) **NO-PCC:** **153** women with (DM-I) | PCC included: • Intensive diabetes management. | **PCC:** **Preterm delivery:** (22/116) **CS delivery:** (73/116) **NO-PCC:** **Preterm delivery:** (41/153) **CS delivery:** (116/153) |
| 7 | **Cyganek** (2016) Poland [64] | **PCC:** **210** women with (DM-I) **NO-PCC:** **313** women with (DM-I) | PCC included: • Glycemic control, • Assessment of diabetes complications. | **PCC:** **First trimester HbA1c:** 6.4 ± SD 1.1 (N = 210) **NO-PCC:** **First trimester HbA1c:** 7.5 ± SD 1.5 (N = 313) |

*(Continued)*

**Table 1.** (Continued)

| Serial No. | Study ID | Participants | Intervention | Outcomes |
|---|---|---|---|---|
| 8 | **Dunne** (1999) United Kingdom[56] | **PCC:** **12** women with (DM-I) **NO-PCC:** **35** women with (DM-I) | PCC included: • Glycemic control, • Assessment of diabetes complications. | **PCC:** **First trimester HbA1c:** 7.9 ± SD 1.4 (N = 12) **Preterm delivery:** (5/12) **CS delivery:** (9/12) **NICU admission:** (2/12) **LGA:** (4/12) **SGA:** (0/12) **Perinatal mortality:** (0/12) **Congenital malformations:** (0/12) **NO-PCC:** **First trimester HbA1c:** 9.6 ± SD 2.4 (N = 35) **Preterm delivery:** (15/35) **CS delivery:** (26/35) **NICU admission:** (12/35) **LGA:** (14/35) **SGA:** (3/35) **Perinatal mortality:** (2/35) **Congenital malformations:** (0/35) |
| 9 | Evers (2004) Netherland [47] | **PCC:** **271** women with (DM-I) **NO-PCC:** **52** women with (DM-I) | PCC included: • Planned pregnancy, • Folic acid supplementation, • Glycemic control. | **PCC:** **First trimester HbA1c:** 6.4 ± SD 0.9 (N = 271) **Congenital malformations:** (11/271) **NO-PCC:** **First trimester HbA1c:** 7.0 ± SD 1.4 (N = 52) **Congenital malformations:** (6/52) |
| 10 | **Fuhrmann** (1986, 1983 & 1984) Germany[27–29] | **PCC:** **185** women with (DM-I) **NO-PCC:** **437** women with (DM-I) | PCC included: • Hospital based glycemic control, • Glucose self-monitoring. | **PCC:** **Congenital malformations:** (2/185) **NO-PCC:** **Congenital malformations:** (31/437) |
| 11 | **Galindo** (2006) Spain[48] | **PCC:** **15** women with (DM-I and DM-II) **NO-PCC:** **111** women with (DM-I and DM-II) | PCC included: • Education, • Glycemic control, • Self-monitoring of blood glucose. | **PCC:** **First trimester HbA1c:** 5.8 ± SD 0.98 (N = 15) **Congenital malformations:** (3/15) **NO-PCC:** **First trimester HbA1c:** 6.6 ± SD 1.72 (N = 111) **Congenital malformations:** (14/111) |
| 12 | **García-Patterso** (1997) Spain[41] | **PCC:** **66** women with (DM-I and DM-II) **NO-PCC:** **119** women with (DM-I and DM-II) | PCC included: • Intensive insulin therapy, • Self-monitoring of blood glucose, • Dietary advice. | **PCC:** **Miscarriage:** (13/66) **CS delivery:** (47/66) **Congenital malformations:** (2/66) **RDS:** (6/66) **Neonatal Hypoglycemia:** (14/66) **Preterm delivery:** (15/66) **Perinatal mortality:** (1/66) **SGA:** (1/54) **NO-PCC:** **Miscarriage:** (15/119) **CS delivery:** (65/119) **Congenital malformations:** (10/119) **RDS:** (12/119) **Neonatal Hypoglycemia:** (30/119) **Preterm delivery:** (29/119) **Perinatal mortality:** (2/119) **SGA:** (9/105) |
| 13 | **Goldman** (1986) Israel [53] | **PCC:** **44** women with (DM-I) **NO-PCC:** **31** women with (DM-I) | PCC included: • Assessment of diabetic complications, • Contraception advice, • Glycemic control and dietary advice. | **PCC:** **First trimester HbA1c:** 7.38 ± SD 0.34 (N = 44) **CS delivery:** (10/44) **Congenital malformations:** (0/44) **Neonatal Hypoglycemia:** (5/44) **RDS:** (1/44) **NO-PCC:** **First trimester HbA1c:** 10.42 ± SD 0.47 (N = 31) **CS delivery:** (13/31) **Congenital malformations:** (3/31) **Neonatal Hypoglycemia:** (8/31) **RDS:** (4/31) |

*(Continued)*

**Table 1.** (Continued)

| Serial No. | Study ID | Participants | Intervention | Outcomes |
|---|---|---|---|---|
| 14 | **Gunton** (2000) Australia [57] | **PCC:** **24** pregnancies (some participants had more than one pregnancy) with (DM-I and DM-II) **NO-PCC:** **69** pregnancies (some participants had more than one pregnancy) with (DM-I and DM-II) **Total N = of women: 61** | PCC included: • Pregnancies planning by optimizing glycemic control before conception (i.e. intensive insulin regimen treatment and tested the blood glucose frequently). | **PCC:** **First trimester HbA1c:** 6.6 ± SD 2.8 (N = 24) **CS delivery:** (3/24) **NO-PCC:** **First trimester HbA1c:** 8.4 ± SD 5.4 (N = 69) **CS delivery:** (33/69) |
| 15 | **Gunton** (2002) Australia [44] | **PCC:** **19** pregnancies (some participants had more than one pregnancy) with (DM-I and DM-II) **NO-PCC:** **16** pregnancies (some participants had more than one pregnancy) with (DM-I and DM-II) **Total Number of women:**31 | PCC included: • Pregnancies planning by optimizing glycemic control before conception | **PCC:** **First trimester HbA1c:** 5.5 ± SD 1 (N = 19) **CS delivery:** (6/19) **LGA:** (5/19) **Congenital malformations:** (0/19) **NO-PCC:** **First trimester HbA1c:** 6.5 ± SD 1.5 (N = 16) **CS delivery:** (8/11) **Congenital malformations:** (1/16) **LGA:** (4/11) |
| 16 | **Heller** (2010) United Kingdom[49] | **PCC:** **99** women with (DM-I) [44 treated with Aspart Insulin 55 women treated with Human Insulin] **NO-PCC:** **223** women with (DM-I) [113 treated with Aspart Insulin 110 women treated with Human Insulin] | PCC included: • Insulin treatment with either Aspart or human insulin. | **PCC:** [x] **First trimester HbA1c:** 6.24 ± SD 0.69 (N = 99) **NO-PCC:** [x] **First trimester HbA1c:** 6.24 ± SD 0.7 (N = 223) |
| 17 | **Hiéronimus** (2004) France [46] | **PCC:** **24** women with (DM-I and DM-II) **NO-PCC:** **36** women with (DM-I and DM-II) | PCC included: Pregnancy programming: - • Pre-conceptional specialized consultation, • Intensification of glycemic self-monitoring, • Optimization of insulin therapy of a preconception HbA1c close to 6%. | **PCC:** **First trimester HbA1c:** 6.79 ± SD 0.72 (N = 24) **Congenital malformations:** (1/24) **NO-PCC:** **First trimester HbA1c:** 8.33 ± SD 1.67 (N = 36) **Congenital malformations:** (8/36) |
| 18 | **Herman** (1999) USA [42] | **PCC:** **24** women with (DM-I) **NO-PCC:** **74** women with (DM-I) | PCC included: • Education and counselling, • Glycemic control, • Assessment of complications of diabetes such as nephropathy and retinopathy. | **PCC:** **Miscarriage:** (4/24) **Congenital malformations:** (1/24) **NO-PCC:** **Miscarriage:** (3/74) **Congenital malformations:** (10/74) |
| 19 | **Holmes** (2017) United Kingdom [51] | **PCC:** **58** women with (DM-I and DM-II) **NO-PCC:** **114** women with (DM-I and DM-II) | PCC included: • Viewing DVD about preconception counselling. | **PCC:** **First trimester HbA1c:** 6.7 ± SD 0.9 (N = 58) **Miscarriage:** (1/58) **CS delivery:** (37/56) **Congenital malformations:** (2/57) **GA at booking(week):** 8.3 ± SD 2.3 (N = 58) **LGA:** (11/57) **Maternal hypoglycemia:** (8/56) **NICU admission:** (15/56) **NO-PCC:** **First trimester HbA1c:** 7.4 ± SD 1.4 (N = 114) **Miscarriage:** (16/114) **CS delivery:** (69/96) **Congenital malformations:** (2/94) **GA at booking(week):** 8.3 ± SD 3.2 (N = 109) **LGA:** (13/93) **Maternal hypoglycemia:** (18/101) **NICU admission:** (30/92) |

(*Continued*)

**Table 1.** (*Continued*)

| Serial No. | Study ID | Participants | Intervention | Outcomes |
|---|---|---|---|---|
| 20 | **Jaffiol** (2000) France [43] | **PCC:** **21** women with (DM-I) **NO-PCC:** **40** women with (DM-I) | PCC included: • Education, • Glycemic control, • Self-monitoring of blood glucose, • Contraception. | **PCC:** **Miscarriage:** (2/21) **CS delivery:** (15/21) **GA at booking(week):** 6.7 ± SD 1.8 (N = 21) **Congenital malformations:** (0/21) **# Perinatal mortality:** (0/21) **RDS:** (2/21) **Neonatal Hypoglycemia:** (1/21) **Preterm delivery:** (7/19) **NO-PCC:** **Miscarriage:** (4/40) **CS delivery:** (21/40) **GA at booking(week):** 11.1 ± SD 5.3 (N = 40) **Congenital malformations:** (3/40) **# Perinatal mortality:** (2/40) **RDS:** (8/40) **Neonatal Hypoglycemia:** (7/40) **Preterm delivery:** (24/34) |
| 21 | **Jensen** (1986) Denmark [68] | **PCC:** **9** women with (DM-I) **NO-PCC:** **11** women with (DM-I) | PCC included: • Glycemic control through continuous subcutaneous insulin infusion and conventional treatment. Initiated two months prior to conception. | **PCC:** **First trimester HbA1c:** 6.9 ± SD 0.2 (N = 9) **NO-PCC:** **First trimester HbA1c:** 7.2 ± SD 0.5 (N = 11) |
| 22 | **Kallas-Koeman** (2012) Canada [60] | **PCC:** **71** women with (DM-I and DM-II) **NO-PCC:** **150** women with (DM-I and DM-II) | PCC included: • Formal PCC at diabetes pregnancy clinics. | **PCC:** **$^x$ First trimester HbA1c:** 6.77 ± SD 0.97 (71) **NO-PCC:** **$^x$ First trimester HbA1c:** 7.63 ± SD 1.69 (N = 150) |
| 23 | **Kekäläinen** (2016) Finland [65] | **PCC:** **96** women with (DM-I) **NO-PCC:** **49** of women with (DM-I) | PCC included: • Pregnancy Planning • Optimizing glycemic control • Medications and screening of other health problems. | **PCC:** **First trimester HbA1c:** 6.76 ± SD 0.82 (N = 96) **Miscarriage:** (15/96) **Preeclampsia:** (18/96) **CS delivery:** (47/96) **Preterm delivery:** (20/96) **Congenital malformations:** (2/96) **LGA:** (35/96) **Shoulder dystocia:** (3/81) **Neonatal hypoglycemia:** (63/96) **Asphyxia:** (4/96) **RDS:** (19/96) **NO-PCC:** **First trimester HbA1c:** 8.30 ± SD 1.14 (N = 49) **Miscarriage:** (14/49) **Preeclampsia:** (10/49) **CS delivery:** (24/49) **Preterm delivery:** (15/49) **Congenital malformations:** (4/49) **LGA:** (14 /49) **Shoulder dystocia:** (3/35) **Neonatal hypoglycemia:** (30/49) **Asphyxia:** (4/49) **RDS:** (9/49) |
| 24 | **Kitzmiller** (1991) USA [38] | **PCC:** **84** women with (DM-I and DM-II) **NO-PCC:** **110** women with (DM-I and DM-II) | PCC included: • Glycemic and dietary control, • Education, • Self-monitoring, • Exercise and contraception. | **PCC:** **Congenital malformations:** (1/84) **NO-PCC:** **Congenital malformations:** (12/110) |
| 25 | **Murphy** (2010) United Kingdom [59] | **PCC:** **181** women with (DM-I and DM-II) **NO-PCC:** **495** women with (DM-I and DM-II) | PCC included: • Glycemic control, • Folic acid supplementation, • Smoking cessation, • Education and preconception counselling. | **PCC:** **Miscarriage:** (28/181) **LGA:** (120/145) **Congenital malformations:** (1/152) **Perinatal mortality:** (1/152) **CS delivery:** (99/181) **Preterm delivery:** (50/150) **SGA:** (7/145) **NO-PCC:** **Miscarriage:** (71/495) **LGA:** (284/372) **Congenital malformations:** (23/408) **Perinatal mortality:** (9/408) **CS delivery:** (222/495) **Preterm delivery:** (116/397) **SGA:** (32/372) |

(*Continued*)

**Table 1.** (Continued)

| Serial No. | Study ID | Participants | Intervention | Outcomes |
|---|---|---|---|---|
| 26 | **Neff** (2014) Ireland[63] | **PCC:** **70** women with (DM-I) **NO-PCC:** **394** women with (DM-I) | PCC included: • Insulin treatment which was continuous subcutaneous infusion and multiple daily injection. | **PCC:** **First trimester HbA1c:** 6.9 ± SD 0.9 (N = 70) **LGA:** (17/70) **SGA:** (4/63) **CS delivery:** (47/70) **Miscarriage:** (7/70) **Preterm delivery:** (11/70) **GA at booking(week):** 6 ± SD 2 (N = 70) **NO-PCC:** **First trimester HbA1c** 7.8 ± SD 1.5 (N = 394) **LGA:** (83/394) **SGA:** (27/331) **CS delivery:** (213/394) **Miscarriage:** (63/394) **Preterm delivery:** (59/394) **GA at booking(week):** 8 ± SD 6 (N = 394) |
| 27 | **Gutaj** (2013) Poland[61] | **PCC:** **43** women with (DM-I and DM-II) **NO-PCC:** **108** women with (DM-I and DM-II) | PCC included: • Pregnancy planning, • Counseling delivered by a diabetes specialist, • Glycemic control by making necessary changes in pharmacotherapy, • Controlling chronic diabetic complications. | **PCC:** [x] **First trimester HbA1c:** 6.15 ± SD 0.82 (N = 43) **NO-PCC:** [x] **First trimester HbA1c:** 8.13 ± SD 01.85 (N = 108) |
| 28 | **Rosenn** (1991) USA[39] | **PCC:** **28** women with (DM-I) **NO-PCC:** **71** women with (DM-I) | PCC included: • Dietary advice • Glycemic control | **PCC:** **First trimester HbA1c:** 8.5 ± SD 0.22 (N = 28) **Congenital malformations:** (0/28) **Miscarriage:** (7/28) **GA at booking(week):** 5.5 ± SD 0.2 (N = 28) **NO-PCC:** **First trimester HbA1c:** 10 ± SD 0.32 (N = 71) **Congenital malformations:** (1/71) **GA at booking(week):** 6.8 ± SD 0.18 (N = 71) **Miscarriage:** (17/71) |
| 29 | **Rowe** (1987) United Kingdom[54] | **PCC:** **14** women with (DM-I) **NO-PCC:** **7** women with (DM-I) | PCC included: • Glycemic control, • Counseling, • Blood glucose self-monitoring. | **PCC:** **First trimester HbA1c:** 9.8 ± SD 2.0 (N = 14) **Congenital malformations:** (0/14) **NO-PCC:** **First trimester HbA1c:** 13.7 ± SD 3.3 (N = 7) **Congenital malformations:** (2/7) |
| 30 | **Steel** (1982 & 1990) United Kingdom[32, 33] | **PCC:** **143** women with (DM-I) **NO-PCC:** **96** women with (DM-I) | PCC included: • Education, • Glycemic control, • Contraception. | **PCC:** **First trimester HbA1c:** 8.4 ± SD 1.3 (N = 143) **Congenital malformations:** (2/143) **Maternal hypoglycemia:** (38/143) **NO-PCC:** **First trimester HbA1c:** 10.5 ± SD 2 (N = 96) **Congenital malformations:** (10/96) **Maternal hypoglycemia:** (8/96) |

*(Continued)*

**Table 1.** (Continued)

| Serial No. | Study ID | Participants | Intervention | Outcomes |
|---|---|---|---|---|
| 31 | **Temple** (2006a & & 2006b) United Kingdom[30, 31] | **PCC:** **110** women with (DM-I) **NO-PCC:** **180** women with (DM-I) | PCC included: • Glycemic control, • Folic acid supplementation, • Smoking cessation, • Education. | **PCC:** **First trimester HbA1c:** 5.9 ± SD 0.9 (N = 110) **GA at booking(week):** 6.6 ± SD 1.8 (N = 110) **Maternal hypoglycemia:** (47/110) **Miscarriage:** (6/110) **Preterm delivery:** (28/110) **Preeclampsia:** (14/110) **CS delivery:** (73/110) **LGA:** (48/110) **Congenital malformations:** (2/110) **# Perinatal mortality:** (1/110) **NO-PCC:** **First trimester HbA1c:** 6.6 ± SD 1.2 (N = 180) **GA at booking(week):** 8.3 ± SD 2.6 (N = 180) **Maternal hypoglycemia:** (65/180) **Preeclampsia:** (22/180) **CS delivery:** (118/180) **Preterm delivery:** (61/180) **Congenital malformations:** (11/180) **Miscarriage:** (22/180) **LGA:** (78/180) **# Perinatal mortality:** (6/180) |
| 32 | **Willhoite** (1993) USA[40] | **PCC:** **62** women with (DM-I and DM-II) **NO-PCC:** **123** women with (DM-I and DM-II) | PCC included: • Counseling by health professional. | **PCC:** **Perinatal mortality:** (4/62) **Congenital malformations:** (1/62) **NO-PCC:** **Perinatal mortality:** (26/123) **Congenital malformations:** (8/123) |
| 33 | **Wong** (2013) United Kingdom[62] | **PCC:** **52** women with (DM-I and DM-II) **NO-PCC:** **109** women with (DM-I and DM-II) | PCC included: • HbA1c monitoring in each trimester, • Insulin treatment, • Pregnancies planning, • Diabetes management by a diabetes (i.e. endocrinologists or general physicians with a special interest in diabetes), • Following up throughout pregnancy with the involvement of dietitians and diabetes educators. | **PCC:** **x First trimester HbA1c:** 7.37 ± SD 1.95 (N = 52) **Congenital malformations:** (1/52) **Perinatal mortality:** (3/52) **NO-PCC:** **x First trimester HbA1c:** 8.33 ± SD 2.33 (N = 109) **Congenital malformations:** (10/109) **Perinatal mortality:** (12/109) |
| 34 | **Wotherspoon** (2017) United Kingdom [52] | **PCC:** **455** women with (DM-I) **NO-PCC:** **292** women with (DM-I) | PCC included: • Pregnancy planning, • Pre-pregnancy counselling (as structured advice about maintaining good blood glucose control and healthy lifestyle (with respect to diet, exercise, BMI, smoking status and alcohol consumption) before trying to become pregnant, including the need to take folate supplements. | **PCC:** **First trimester HbA1c:** 7.0 ± SD 0.8 (N = 455) **Pre-eclampsia:** (74/448) **CS delivery:** (307/454) **Perinatal mortality:** (12/449) **SGA:** (26/446) **LGA:** (230/446) **Congenital malformations:** (15/454) **NICU admission:** (218/436) **NO-PCC:** **First trimester HbA1c:** 7.5 ± SD 1.1 (N = 292) **Pre-eclampsia:** (49/286) **CS delivery:** (200/286) **Perinatal mortality:** (6/284) **SGA:** (31/284) **LGA:** (149/284) **Congenital malformations:** (13/291) **NICU admission:** (178/277) |

**DM-I:** Diabetes Mellitus type I, **DM-II:** Diabetes Mellitus type II, **GA:** Gestational Age, **GWG:** Gestational Weight Gain, **HbA1c:** Glycated Haemoglobin, **LGA:** Large for Gestational Age, **NICU:** Neonatal Intensive Care Unit, **NO-PCC:** No Preconception Care, **PCC:** Preconception Care, **RDS:** Respiratory Distress Syndrome, **SGA:** Small for Gestational Age

x Calculated mean

~ LGA and macrosomia

# sum of stillbirth and neonatal death

**Table 2. Characteristics of included case control study.**

| Study ID | Participants | Intervention vs. comparison | Outcomes |
|---|---|---|---|
| **Garcia- Ingelmo** 1998 Spain [67] | **PCC:**12 **NO-PCC:**12 | **PCC included:** • Glycaemic control | **PCC:** **First trimester HbA1c:** 6.7±0.58 **Congenital malformations:** 3/12 **Macrosomia:** 6/12 **NO-PCC:** **First trimester HbA1c:** 8.29±1.32 **Congenital malformations:** 2/12 **Macrosomia:** 4/12 |

**NO-PCC:** No preconception care, **PCC:** pre-conception care, **HbA1c:** Glycated Haemoglobin.

pregnancies in the trial [66]. Most of studies did not report the differences in the outcomes among type 1 versus type 2 diabetes, subsequently, we could not conduct the analysis separately for each type of diabetes.

**Interventions.** The PPC in all the cohort studies included control and self-monitoring of blood glucose except for one study which was designed to examine the effectiveness of pre-pregnancy counseling on perinatal outcomes [40]. In addition to glycemic control, ten studies included screening and treatment of complications of diabetes in the PPC program [29, 42, 45, 50, 53, 55, 56, 58, 61, 62]. Eleven studies (12 reports) reported comprehensive PCC program including control and self-monitoring of blood glucose in addition to any combination of the following: folic acid supplementation, diet and exercise, smoking cessation, alcohol withdrawal advice, and discontinuation of teratogenic drugs [31, 38, 47, 50–52, 58–61, 63].

**Outcomes measure.** In this review, a total of 14 outcomes were reported in the cohort studies, including five maternal outcomes: HbA1c in the first trimester, CS delivery, miscarriage, GA at first antenatal booking, and maternal hypoglycemia during the first trimester. There were nine neonatal outcomes (congenital malformations, perinatal mortality, preterm

**Table 3. Characteristics of included RCT studies.**

| Study ID | Participants | Intervention vs. comparison | Outcomes |
|---|---|---|---|
| **DCCT Research Group** 1996 USA [66] | 94 women with 135 pregnancies in the intensive treatment group and 86 women with 135 pregnancies in the conventional treatment group | Intensive glycemic control (IGC) (multiple daily insulin injections or continuous infusion pump and self-monitoring) | **PCC:** **At conception HbA1c:** 7.4± SD 1.3 (N = 132) **Congenital malformations:** (1/135) **Spontaneous abortion:**(18/ 135) **Perinatal mortality** (Intrauterine deaths) (5/135) **NO-PCC:** **At conception HbA1c:** 8.1 ±SD 1.7 (N = 135) **Congenital malformations:** (8/135) **Spontaneous abortion:** (14/ 135) **Perinatal mortality** (Intrauterine deaths) (9/ 135) |

**NO-PCC:** No preconception care, **PCC:** pre-conception care, **HbA1c:** Glycated Haemoglobin.

Table 4. Risk of bias assessment of the included studies.

| Study | Selection | | | | Comparability | | Outcome | | | Total | Risk of Bias assessment/Notes |
|---|---|---|---|---|---|---|---|---|---|---|---|
| | Exposed | Control | Exposure | Outcome | Age Match | Other | Method Assess | Adequate Follow | Complete Follow-Up | | |
| **Boulot 2003[45]** | * | * | * | * | * | * | * | * | * | 9 | **Low risk** |
| **Cousins 1991 [37]** | * | * | * | * | unclear | unclear | * | * | unclear | 6 | **High risk** No comparability |
| **Damm 1989[36]** | * | * | * | * | unclear | unclear | unclear | * | | 5 | **High risk** Unclear group comparability |
| **Dicker 1988[55]** | | * | * | * | * | * | * | * | * | 8 | **Low risk** |
| **Egan 2016[50]** | | * | * | * | * | * | * | * | * | 8 | **Low risk** |
| **Cyganek 2010 [58]** | * | * | * | * | * | * | * | * | * | 9 | **Low risk** |
| **Cyganek 2016 [64]** | * | * | * | * | * | | * | * | * | 8 | **Low risk** |
| **Dunne 1999 [56]** | * | * | * | * | | | * | * | * | 7 | **High risk** The study was an audit, groups were different in smoking, no statistical adjustment done. |
| **Evers 2004[47]** | * | * | * | * | Unclear | Unclear | * | * | * | 7 | **High risk** Confounding factors such as smoking, education level and social class were not examined. The results of HbA1c during the first trimester were not available for 29% of the whole study group |
| **Fuhrmann 1983, 1984, 1986[27–29]** | | * | * | * | Unclear | Unclear | * | * | | 5 | **High risk** no description of the possible confounding factors or adjustment |
| **Galindo 2006 [48]** | * | * | * | * | unclear | unclear | * | * | * | 7 | **High risk** It is unclear if factors influencing the outcome were similar in both groups, no statistical adjustment was done |
| **García-Patterson 1997 [41]** | | * | * | * | * | unclear | * | * | * | 7 | **Low risk** |
| **Goldman 1986 [53]** | | * | * | * | * | Unclear | * | * | * | 7 | **High risk** Difference in smoking and BMI between the groups not assessed |
| **Gunton 2000 [57]** | | * | * | * | * | * | * | * | | 7 | **Low risk** |
| **Gunton 2002 [44]** | * | * | | * | Unclear | Unclear | * | * | * | 6 | **High risk** Difference in the duration of diabetes between the groups not controlled for |
| **Gutaj 2013[61]** | | * | * | | * | * | * | * | * | 7 | **Low risk** |
| **Heller 2010 [49]** | * | * | * | * | * | * | * | * | * | 9 | **Low risk** |
| **Hiéronimus 2004[46]** | | * | * | * | Unclear | Unclear | * | * | * | 6 | **High risk** no description of the possible confounding factors or adjustment |
| **Herman 1999 [42]** | | * | * | * | * | Unclear | * | * | | 6 | **High risk** The groups are different in duration of diabetes other confounders not addressed, no adjustment |
| **Holmes 2017[51]** | | | * | * | * | * | * | * | * | 7 | **Low risk** |

*(Continued)*

**Table 4.** (Continued)

| Study | Selection | | | | Comparability | | Outcome | | | Total | Risk of Bias assessment/Notes |
|---|---|---|---|---|---|---|---|---|---|---|---|
| | Exposed | Control | Exposure | Outcome | Age Match | Other | Method Assess | Adequate Follow | Complete Follow-Up | | |
| Jaffiol 2000[43] | | * | * | * | * | * | * | * | * | 8 | **Low risk** |
| Jensen 1986[68] | | * | | * | Unclear | Unclear | * | * | | 4 | **High risk** Differences in the severity of diabetes, five of the 11 control women were treated in the diabetic clinic in the hospital before pregnancy so they knew about the importance of glycemic control |
| Kallas-Koeman 2012[60] | * | * | * | * | * | * | * | * | * | 9 | **Low risk** |
| Kekäläinen 2016 [65] | | * | * | * | * | * | * | * | * | 8 | **Low risk** |
| Kitzmiller 1991 [38] | | * | * | * | Unclear | Unclear | * | * | * | 6 | **High risk** Unclear if there is difference between the groups. |
| Murphy 2010 [59] | * | * | * | * | * | * | * | * | * | 9 | **Low risk** |
| Neff 2014[63] | * | * | * | * | * | * | * | * | | 8 | **Low risk** |
| Rosenn 1991[39] | | * | * | * | Unclear | Unclear | * | * | | 5 | **High risk** 52% lost to follow up, different baseline characteristics including duration of diabetes, age, complications of diabetes |
| Rowe 1987 [54] | | * | | * | Unclear | Unclear | | * | * | 4 | **High risk** no description of the possible confounding factors or adjustment |
| Steel 1982, 1990 [32, 33] | * | * | * | * | * | | * | * | * | 7 | **Low risk** There is no significant clinical age difference between the groups. However, there is different number of smokers. No regression analysis was done to address this difference |
| Temple a & b 2006 [30, 31] | * | * | * | * | * | * | * | * | * | 9 | **Low risk** |
| Willhoite 1993 [40] | * | * | unclear | * | | | | * | * | 5 | **High risk** Base line characteristics of the two groups were significantly different in age, duration of diabetes and smoking. The two groups did not receive the same antenatal intra-partum and postnatal care. |
| Wong 2013 [62] | | * | * | * | * | * | * | * | * | 8 | **Low risk** |
| Wotherspoon 2017[52] | * | * | | * | * | * | * | * | * | 8 | **Low risk** |

Risk of bias was assessed using the Newcastle-Ottawa Scale (NOS). The number of stars represents the risk of bias; the maximum number of stars is nine, studies were classified as "low risk of bias" if they received a score of six stars or more, along with at least one star in the comparability domain. Studies at "high risk of bias" scored less than six stars or scored no stars in comparability domain, irrespective of the number of stars scored.

delivery, macrosomia/LGA, SGA, neonatal hypoglycemia, admission to NICU, RDS and shoulder dystocia). The most frequently reported outcomes were HbA1c in the first trimester (24 studies) [31, 32, 36, 39, 44, 46–57, 60–65, 68] and congenital malformations (25 studies) [29, 31, 32, 36–48, 50–54, 56, 59, 62, 65]. Fourteen studies examined the reduction in CS

delivery in women who received PCC compared to those who did not [31, 41, 43, 44, 50–53, 56–59, 63, 65]. Eleven studies compared the miscarriage rate between the two groups [31, 39, 41–43, 50, 51, 55, 59, 63, 65].

Perinatal mortality [31, 40, 41, 43, 45, 50, 52, 56, 59, 62] and preterm delivery [31, 41, 43, 50, 56, 58, 59, 63, 65] were reported in ten and nine studies respectively, while LGA, macrosomia [31, 44, 50–52, 56, 59, 63, 65] SGA [41, 50, 52, 56, 59, 63], and neonatal hypoglycemia [41, 43, 50, 53, 65] were reported in nine, six and five studies respectively. Admission to NICU [50–52, 56], and neonatal RDS [41, 43, 53, 65] were reported in four studies. The least reported outcomes were maternal hypoglycemia in three studies (all reported results among type 1 diabetes) [31, 33, 51] and shoulder dystocia in two studies [50, 65].

**Assessment of the methodological quality of the included studies.** We used the NOS Form for Cohort Studies to determine the level of bias of cohort studies included in this review (Table 4). 21 studies were determined to be at a low risk of bias [30–33, 41, 43, 45, 49–52, 55, 57–65], while 15 studies were judged to be at a high risk of bias [27–29, 36–40, 42, 44, 46–48, 53, 54, 56, 68]. Some of the studies at a high risk of bias were initially designed to assess aspects of PPC other than its effectiveness in improving maternal and perinatal outcomes, hence the poor methodological design when assessed with the NOS [40, 42, 56].

The cohort studies included in this review (Table 1) had adequate description of participants including comparison between the PPC group and the control group regarding some confounding factors such as the duration of diabetes and frequency of renal and vascular complications. However, most studies did not address the effect of the confounding factors on the outcomes except for ten studies (11 reports) which used regression analysis to evaluate the independent effect of the PPC [31, 45, 51, 52, 57–60, 63, 65]. In most cohort studies, blinding of the control group was adequate because they were recruited after the inception of pregnancy while attending the antenatal care. In four studies, [33, 38, 49, 50] blinding was inadequate as the control groups were invited to participate in the PCC program and hence were informed about it. All participants received the same antenatal and postnatal care except for six studies [42, 47, 49–52] where participants were followed up in different health settings. In all cohort studies, the compliance of participants to follow up was adequate except for Rosenn et al [39] where 52% of the PCC group were lost to follow-up, and Jensen et al [68] where 3 out of 11 participants in control group did not comply with study protocol as they rejected the self-glucose monitoring. The assessors of the outcomes were not blinded to the participants' allocation; however, we do not believe this would have introduced bias due to the objective nature of the outcomes in this review.

One case control study was included in this review [67]. The study encompassed a small sample size which included 12 cases each for cases and control. Both recall bias and detection bias cannot be excluded.

One trial was included in this review [66] (Table 3). The design of the trial was not clear; neither the method of randomization nor the allocation concealment was described. In addition to that, lack of blinding introduced bias because both groups were aware of the importance of the glycemic control and the complications of diabetes during pregnancy.

**Effects of intervention.** Fourteen outcomes were identified after examining all the studies; meta-analysis was possible for 34 cohort studies with 8199 participants.

*Gestational age at booking for antenatal care.* The results of the meta-analysis on the effect of PCC on the early booking for antenatal care showed that women who attended PCC booked approximately ten days earlier for antenatal care (MD 1.31; 95% CI: 1.40–1.23); five studies, 1081 women very low-certainty evidence)) (Fig 2) (Table 5). The quality of evidence was downgraded from low-grade (observational study) to very low-grade due to the high risk of bias in the study with the largest weight [39] and high unexplained heterogeneity (Table 6).

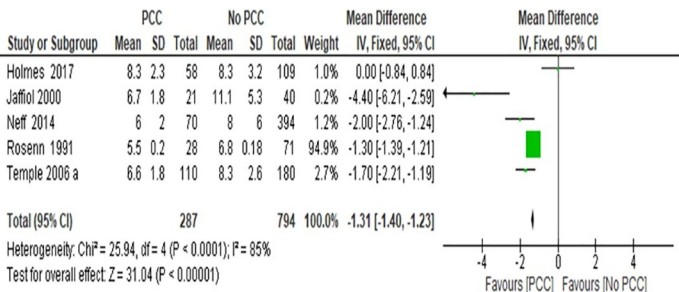

The large green square represents the estimate effect of the study with the highest weight and very precise 95% CI. The black diamond represents the pooled difference estimate. Heterogeneity is quantified by I² statistics, an I² value ≥ 50 indicates substantial heterogeneity. Estimated results are presented as mean difference with 95% Confidence Interval. PCC= Preconception care; No PCC=No preconception care; CI= Confidence intervals.

**Fig 2. The mean gestational age at the time of the first antenatal visit from five studies of women with pre-existing diabetes mellitus who did or did not receive preconception care.**

*Congenital malformations.* The result of the meta-analysis on the effect of PCC on congenital malformations suggested that PCC resulted in a large reduction in congenital malformations (RR 0.29; 95% CI: 0.21–0.40, 25 studies; 5903 women; high-certainty evidence) (Fig 3) (Table 5). We considered the body of evidence for the effect of PCC on the reduction of congenital malformations to be high quality mainly due to the large effect size with precise and narrow confidence intervals, consistency of direction of effect throughout most of the included studies, no indirectness of evidence, and no heterogeneity or publication bias. Bias in the included studies was considered moderate (59% of the participants were from studies at low-risk of bias) (Fig 3, Table 6).

*HbA1c.* Meta-analysis of 24 studies which reported HbA1c showed that PCC likely results in a reduction in HbA1c in the first trimester of pregnancy by an average of 1.27%; (MD 1.27; 95% CI: 1.33–1.22; 4927 women; moderate-certainty evidence) (Fig 4) (Tables 5 & 6). We considered bias in the included studies low (77% of the participants were from studies at low risk of bias) (Fig 4, Table 6), while heterogeneity can be explained by long span of time between the

**Table 5. Pooled estimates effect of preconception care.**

| Outcome | No of Studies | No of Participants | Effect estimate Risk Ratio [95% CI] |
|---|---|---|---|
| Congenital malformations | 25 | 5903 | 0.29 [0.21, 0.40] |
| Maternal hypoglycemia | 3 | 686 | 1.38 [1.07, 1.79] |
| Preterm delivery | 9 | 2414 | 0.85 [0.73, 0.99] |
| Perinatal mortality | 10 | 3071 | 0.46 [0.30, 0.73] |
| Small for gestational age | 6 | 2261 | 0.52 [0.37, 0.75] |
| Admission to neonatal intensive care unit | 4 | 1322 | 0.75 [0.67, 0.84] |
| Cesarean section delivery | 14 | 3641 | 1.02 [0.96, 1.07] |
| Miscarriage | 11 | 2698 | 0.86 [0.70, 1.06] |
| Large for gestational age / macrosomia | 9 | 2787 | 1.06 [0.97, 1.15] |
| Neonatal hypoglycemia | 5 | 880 | 0.93 [0.74, 1.18] |
| Respiratory distress syndrome | 4 | 466 | 0.78 [0.47, 1.29] |
| Shoulder dystocia | 2 | 530 | 0.28 [0.07, 1.12] |
| | | | **Mean Difference [95% CI]** |
| Gestational age at booking for antenatal care | 5 | 1081 | -1.31 [-1.40, -1.23] |
| HbA1c in the first trimester | 24 | 4927 | -1.27 [-1.33, -1.22] |

**CI**: Confidence Interval.

**Table 6. Summary of findings table.**

[Preconception care] compared to [no preconception care] or [routine care] for [improving maternal and perinatal outcomes]

**Patient or population**: [improving maternal and perinatal outcomes]
**Setting**: Hospital setting
**Intervention**: [Preconception care]
**Comparison**: [no preconception care] or [routine care]

| Outcomes | Anticipated absolute effects[*] (95% CI) | | Relative effect (95% CI) | № of participants (studies) | Certainty of the evidence (GRADE) | Comments |
|---|---|---|---|---|---|---|
| | Risk with [no preconception care] or [routine care] | Risk with [Preconception care] | | | | |
| Congenital malformations follow up: mean 9 months [a] | 70 per 1,000 | **20 per 1,000** (15 to 28) | **RR 0.29** (0.21 to 0.40) | 5903 (25 observational studies) | ⊕⊕⊕⊕ HIGH | [Preconception care] results in large reduction in congenital malformations. |
| Perinatal mortality follow up: mean 9 months [b] | 46 per 1,000 | **21 per 1,000** (13 to 33) | **RR 0.46** (0.30 to 0.73) | 3071 (10 observational studies) | ⊕⊕⊕○ MODERATE | [Preconception care] results in large reduction in perinatal mortality. |
| Gestational age at booking follow up: mean 9 months [c] | The mean gestational age at booking was **8.5 Weeks** | mean **1.31 Weeks fewer** (1.4 fewer to 1.23 fewer) | - | 1081 (5 observational studies) | ⊕○○○ VERY LOW | [Preconception care] may result in a slight reduction in gestational age at booking. |
| Hemoglobin A1c (HbA1c) follow up: mean 9 months [d] | The mean hemoglobin A1c was **8.3%** | mean **1.32% lower** (1.34 lower to 1.23 lower) | - | 4927 (24 observational studies) | ⊕⊕⊕○ MODERATE | [Preconception care] likely results in a reduction in HbA1c. |
| Maternal hypoglycemia follow up: mean 9 months [e] | 241 per 1,000 | **333 per 1,000** (258 to 432) | **RR 1.38** (1.07 to 1.79) | 686 (3 observational studies) | ⊕○○○ VERY LOW | [Preconception care] has no effect on Maternal hypoglycemia |
| Preterm delivery follow up: mean 9 months [f] | 250 per 1,000 | **213 per 1,000** (183 to 248) | **RR 0.85** (0.73 to 0.99) | 2414 (9 observational studies) | ⊕⊕⊕○ MODERATE | [Preconception care] likely results in a slight reduction in preterm delivery. |
| Small for gestational age follow up: mean 9 months [g] | 88 per 1,000 | **46 per 1,000** (32 to 66) | **RR 0.52** (0.37 to 0.75) | 2261 (6 observational studies) | ⊕⊕⊕○ MODERATE | [Preconception care] reduces small for gestational age. |

[*]**The risk in the intervention group** (and its 95% confidence interval) is based on the assumed risk in the comparison group and the **relative effect** of the intervention (and its 95% CI).

CI = Confidence interval; RR = Risk ratio

**GRADE Working Group grades of evidence**

**High certainty:** We are very confident that the true effect lies close to that of the estimate of the effect

**Moderate certainty:** We are moderately confident in the effect estimate: The true effect is likely to be close to the estimate of the effect, but there is a possibility that it is substantially different

**Low certainty:** Our confidence in the effect estimate is limited: The true effect may be substantially different from the estimate of the effect

**Very low certainty:** We have very little confidence in the effect estimate: The true effect is likely to be substantially different from the estimate of effect

[a] Upgraded to high because of large effect size, consistency of direction of effect, no indirectness of evidence, and no heterogeneity or publication bias.

[b] Upgraded to moderate due to the narrow confidence intervals, consistency of direction of effect, no indirectness of evidence, and low risk of bias, no heterogeneity or publication bias.

[c] Downgraded to very low-grade due to the high risk of bias in the study with the largest weight [39] and high unexplained heterogeneity

[d] Upgraded to moderate-certainty level because of low bias (77% of the participants were from studies at low risk of bias), while heterogeneity can be explained by long span of time between the first and the last study (1982 and 2017), The publication bias can be explained with the heterogeneity.

[e] Downgraded to very low-level certainty because, inconsistency, low bias and high heterogeneity

[f] Upgraded to moderate-certainty level because of narrow confidence intervals, consistency of direction of effect, no indirectness of evidence, low risk of bias, low heterogeneity, no evidence of selective reporting.

[g] Upgraded to moderate-certainty level because the large effect size with precise narrow confidence interval, consistency of direction of effect, no indirectness of evidence, and no heterogeneity and no evidence of selective reporting.

first and the last study (1982 and 2017), during which time many innovations in the management of diabetes has occurred with substantial reduction in the target level of HbA1c. The

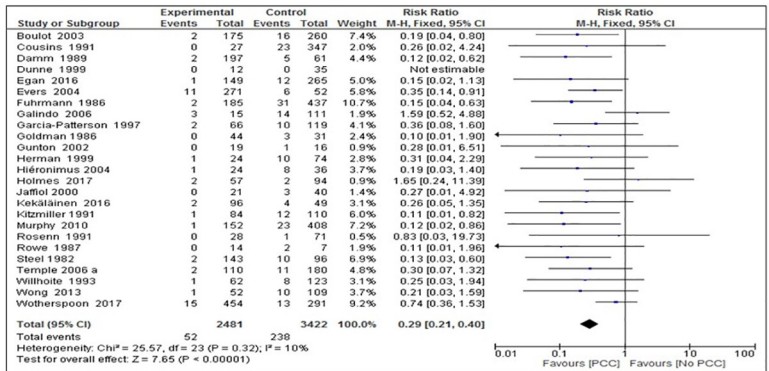

The black diamond represents the pooled risk estimate. Heterogeneity is quantified by I² statistics, an I² value ≥ 50 indicates substantial heterogeneity. Estimated results are presented as risk ratio with 95% Confidence Interval. PCC= Preconception care; No PCC= No preconception care; CI= Confidence intervals.

**Fig 3. Risk ratio for congenital malformations from 25 studies of women with pre-existing diabetes mellitus who did or did not receive preconception care.**

apparent publication bias in this outcome can be explained with the heterogeneity associated with this analysis.

*Maternal hypoglycemia*. We are uncertain about the effect of PCC on maternal hypoglycemia during the first trimester of pregnancy; (RR 1.38; 95% CI: 1.07–1.79); three studies; 686 women; very low-certainty evidence) (Fig 5) (Table 5). The grade of evidence was downgraded from low to very low due to inconsistency of the direction of effect and high heterogeneity (I² = 76%) in the included studies (Table 6). The true effect is likely to be substantially different from the effect estimated in this review.

*Preterm delivery*. The results of the meta-analysis on the effect of PCC on preterm delivery indicate that PCC lead to a slight reduction in preterm delivery rate among women with diabetes (RR 0.85; 95% CI: 0.73–0.99; nine studies, 2414 women; moderate-certainty evidence) (Fig 6) (Table 5). We upgraded the body of evidence for the effect of PCC on the reduction of preterm delivery to moderate quality. This upgrade was based on the narrow confidence intervals around the point estimate, consistency of direction of effect in most of the included studies, no indirectness of evidence, low risk of bias of the body of evidence as only 1.9% of the participants were from one study with high risk of bias. The low heterogeneity with no evidence of

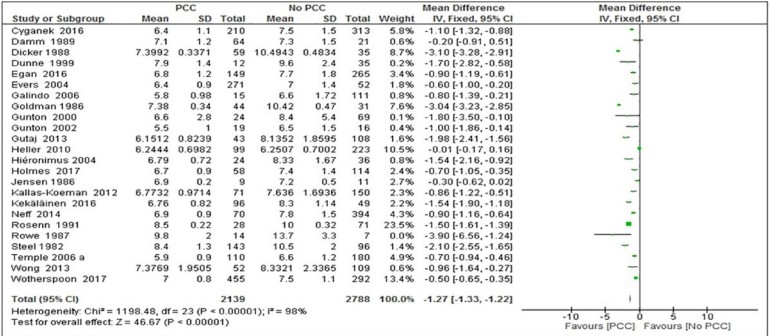

The black diamond represents the pooled difference estimate. Heterogeneity is quantified by I² statistics, an I² value ≥ 50 indicates substantial heterogeneity. Estimated results are presented as mean difference with 95% Confidence Interval. PCC= Preconception care; No PCC= No preconception care; CI= Confidence intervals.

**Fig 4. First trimester means values of glycosylated hemoglobin (HbA1c) from 24 studies of women with pre-existing diabetes mellitus who did or did not receive preconception care.**

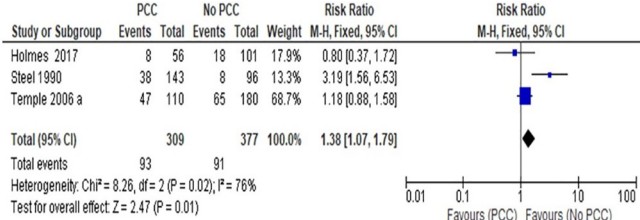

The large blue square represents the estimate effect of the study with the highest weight and very precise 95% CI. The black diamond represents the pooled risk estimate. Heterogeneity is quantified by I² statistics, an I² value ≥ 50 indicates substantial heterogeneity. Estimated results are presented as risk ratio with 95% Confidence Interval. PCC= Preconception care; No PCC= No preconception care; CI= Confidence intervals.

**Fig 5. Risk ratio for maternal hypoglycaemia from three studies of women with pre-existing diabetes mellitus who did or did not receive preconception care.**

selective reporting increase our confidence in the outcome of a small reduction in preterm delivery (Fig 6, Table 6)

*Perinatal mortality.* The meta-analysis results on the effect of PCC on perinatal mortality indicates that PCC results in a large reduction in perinatal mortality (RR 0.46; 95% CI: 0.30–0.73; ten studies; 3071 women; moderate-certainty evidence) (Fig 7) (Table 5). The quality of evidence has been upgraded to moderate due to the narrow confidence intervals, consistency of direction of effect in most of the included studies, no indirectness of evidence, and low risk of bias of the body of evidence as only 7.6% of the participants were from two studies at high risk of bias and no heterogeneity or publication bias.

*Small for gestational age.* The result of the meta-analysis indicates that PCC may result in large reduction in SGA (RR 0.52; 95% CI: 0.37–0.75; six studies, 2261 women; moderate-certainty evidence) (Fig 8) (Table 5). We upgraded the body of evidence for the effect of PCC on the reduction of SGA to moderate-quality on account of the large effect size (48% reduction in SGA) with precise narrow confidence interval, consistency of direction of effect throughout the included studies, no indirectness of evidence, and no heterogeneity and no evidence of selective reporting. We considered bias in the included studies as low (2% of the participants were from one study at high risk of bias) (Fig 8, Table 6).

*NICU admission.* The result of the meta-analysis on the effect of PCC on admission to NICU indicates that PCC may reduce the rate of NICU admissions (RR 0.75; 95% CI: 0.67–0.84; four studies; 1322 women; moderate-certainty evidence) (Fig 9) (Table 5). The body of evidence was upgraded owing to precise narrow confidence intervals, consistency of direction of effect, no indirectness of evidence, and no heterogeneity and no evidence of selective reporting. We considered bias in the included studies as low.

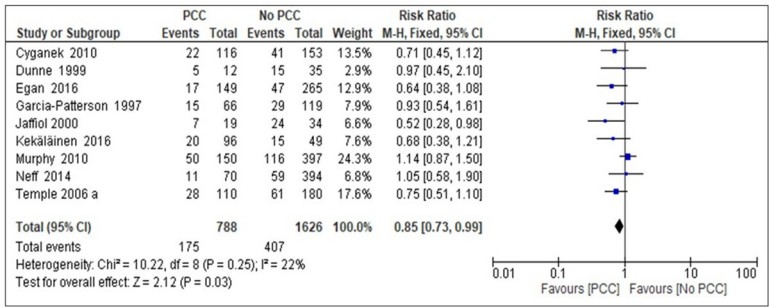

The black diamond represents the pooled risk estimate. Heterogeneity is quantified by I² statistics, an I² value ≥ 50 indicates substantial heterogeneity. Estimated results are presented as risk ratio with 95%. PCC= Preconception care; No PCC= No preconception care; CI= Confidence intervals.

**Fig 6. Risk ratio for preterm delivery from nine studies of women with pre-existing diabetes mellitus who did or did not receive preconception care.**

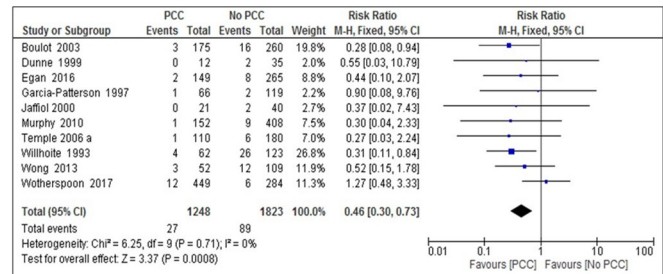

The black diamond represents the pooled risk estimate. Heterogeneity is quantified by I² statistics, an I² value ≥ 50 indicates substantial heterogeneity. Estimated results are presented as risk ratio with 95% Confidence Interval. PCC= Preconception care; No PCC= No preconception care; CI= Confidence intervals.

**Fig 7. Risk ratio for perinatal mortality from ten studies of women with pre-existing diabetes mellitus who did or did not receive preconception care.**

**Other outcomes.** Meta-analysis showed that the PCC may have little or no effect in reducing the CS rate (RR 1.02; 95% CI: 0.96–1.07; 14 studies; 3641 women; low-certainty evidence), miscarriage rate (RR 0.86; 95% CI: 0.70–1.06; 11 studies; 2698 women; low- certainty evidence), macrosomia rate, (RR 1.06; 95% CI: 0.97–1.15; nine studies; 2787 women, low- certainty evidence), neonatal hypoglycemia (RR 0.93; 95% CI: 0.74–1.18; five studies; 880 women; low- certainty evidence), RDS (RR 0.78; 95% CI: 0.47–1.29; four studies; 466 women; very low-certainty evidence) and shoulder dystocia (RR 0.28; 95% CI: 0.07–1.12; 2 studies; 530 women; very low- certainty evidence).

**Results of sensitivity analysis.** We performed sensitivity analysis by excluding studies with high risk of bias from the meta-analysis. Overall, results and conclusions were not changed (Figs 1–7, S3 File).

*Publication bias.* We examined the possibility of publication bias by evaluating the asymmetry of the Funnel Plots (Fig 10). Analysis of the effect of PCC on congenital malformations and perinatal mortality (Fig 10A & 10B) demonstrated symmetrical distribution of the studies which can reasonably exclude publication bias. Analysis of the effect on HbA1C, showed asymmetrical distribution of the studies (Fig 10C), however this can be explained by the marked heterogeneity associated with this outcome [69]. We found no evidence of selective reporting of outcomes in all included studies.

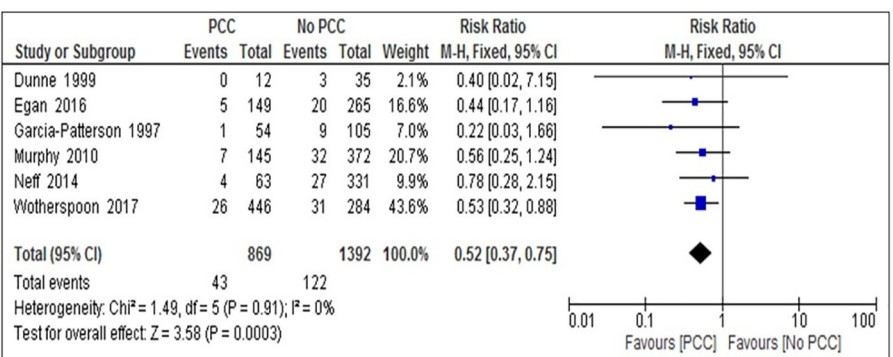

The black diamond represents the pooled risk estimate. Heterogeneity is quantified by I² statistics, an I² value ≥ 50 indicates substantial heterogeneity. Estimated results are presented as risk ratio with 95% Confidence Interval. PCC= Preconception care; No PCC= No preconception care; CI= Confidence intervals.

**Fig 8. Risk ratio for small for gestational age from six studies of women with pre-existing diabetes mellitus who did or did not receive preconception care.**

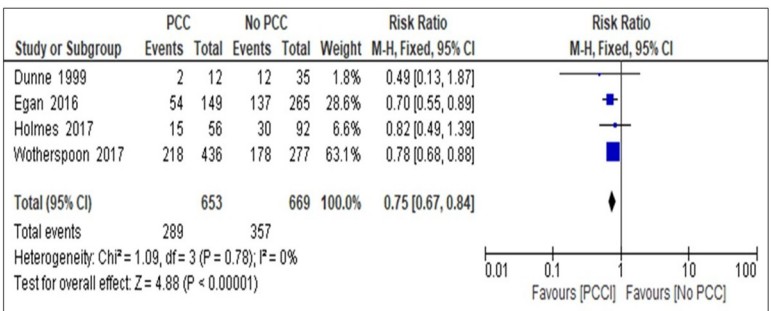

The large blue square represents the estimate effect of the study with the highest weight and very precise 95% CI. The black diamond represents the pooled risk estimate. Heterogeneity is quantified by I² statistics, an I² value ≥ 50 indicates substantial heterogeneity. Estimated results are presented as risk ratio with 95% Confidence Interval. PCC= Preconception care; No PCC= No preconception care; CI= Confidence intervals.

**Fig 9. Risk ratio for neonatal intensive care unit admission from four studies of women with pre-existing diabetes mellitus who did or did not receive preconception care.**

## Discussion

The results of this review showed that PCC for mothers with pregestational diabetes is effective in improving the outcomes for several maternal and neonatal complications associated with pregestational diabetes. PCC results in a large reduction in congenital malformations. It probably results in a reduction of HbA1c in the first trimester of pregnancy, perinatal mortality and in slightly earlier booking of mothers for antenatal care. PCC is likely to result in a slight reduction of preterm birth rate. We are uncertain about the effect of PCC on maternal hypoglycemia during the first trimester of pregnancy.

Congenital malformations are one of the principal contributors to the high perinatal mortality observed in pregnancies complicated with pregestational diabetes [7, 8, 70, 71]. Maternal hyperglycemia at the time of organogenesis is the main teratogen [72, 73]. Evidence from clinical and experimental studies showed that hyperglycemia leads to the production of reactive oxygen species and depletion of antioxidants, which in turn causes intracellular oxidative stress, cell injury and cell death at the time of organogenesis [74, 75]. It is not surprising that PCC has a large effect on reducing the rate of congenital malformations as it provides the right window of opportunity for optimum control of hyperglycemia before the early critical weeks of conception and organogenesis. Another intervention with proven effectiveness in the prevention of congenital malformations in this high-risk group is preconception folic acid supplementation [76, 77]. Folic acid supplementation was part of almost all PCC interventions of the studies included in this review and may have contributed to the large effect of PCC in reducing congenital malformations rate.

The results of this review showed that women who received PCC achieved better control of hyperglycemia compared to those who didn't attend PCC as evident by the significantly lower mean HbA1c level of the intervention group compared to the control group. Many studies confirmed the incremental increase in the rate of adverse pregnancy outcomes, among women with diabetes, with the increase in the level of HbA1c [78–81] and the significant risk reduction in congenital malformations with one percent reduction in HbA1c level [82].

The etiology of preterm birth is complex and many medical, socioeconomic, and psychological factors interplay in the causation of preterm delivery [83]. Nevertheless, mothers with diabetes are at greater risk of preterm birth compared to the background population [6].

A recently published systematic review and modelling analysis of the global estimate of the rate of preterm birth estimated that 14.84 million preterm live births were born in the year 2014, with the majority born in low- and middle-income countries [84]. Preterm birth is the

**a**

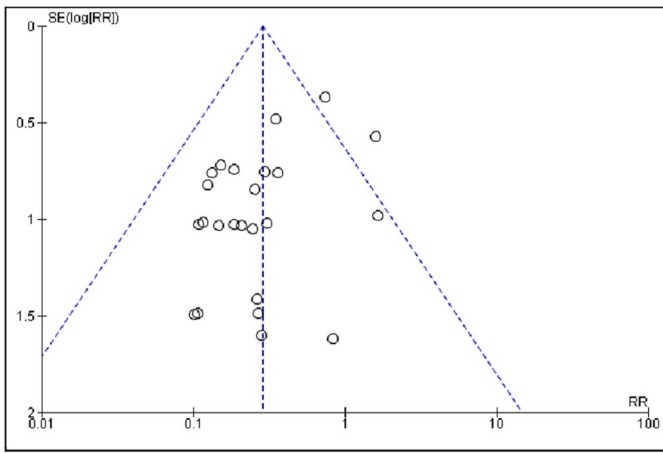

**b**

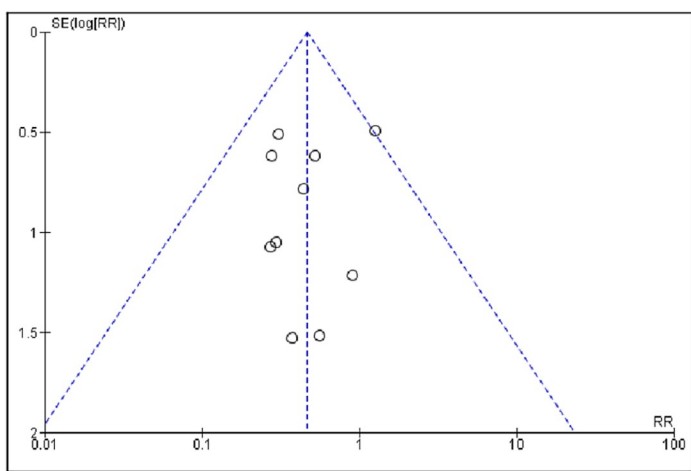

**c**

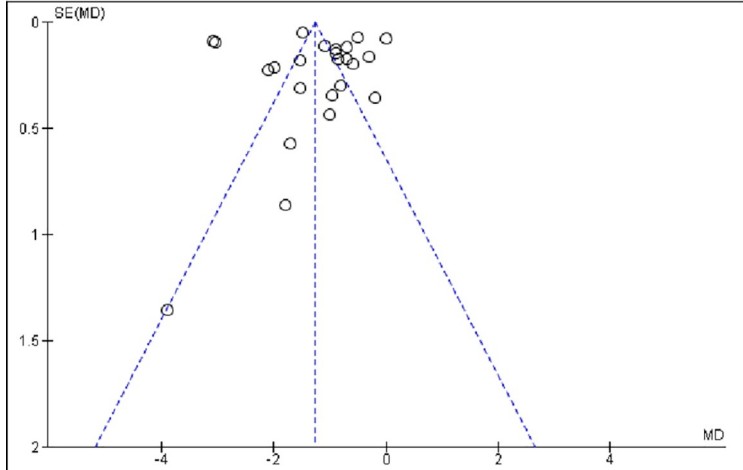

**Fig 10.** Funnel plots for studies included in the meta-analysis for the effect of PCC on congenital anomalies (a), perinatal mortality (b) and HbA1c (c).

leading cause of under-five mortality and one of the main causes of both short and long-term morbidity [85, 86]. In addition, preterm birth is associated with high cost to the health system and the families of the preterm born infant [87]. Based on the above account, the reduction in the prevalence of preterm birth achieved by the attendance of PCC will have a considerable impact on the perinatal mortality and neonatal and infant morbidity for children born to mothers with pregestational diabetes. This assertion has been confirmed by the marked reduction in perinatal mortality and in the admission to NICU for infants of mothers who received PCC compared to those who did not in this review.

The results of this review showed a significant reduction in the rate of SGA in women who attended PCC. This effect maybe secondary to the significant reduction in congenital malformations rate. The association between SGA and congenital malformations, especially those for the heart and the urinary tract, has been documented in published reports [88, 89]. Nevertheless, we cannot exclude the effect of healthy lifestyle promotion including smoking cessation, weight control, and teratogenic drugs avoidance as part of many PCC programs, which are factors contributing to the reduction in the rate of SGA [90].

The effects of PCC on the reduction of congenital malformations, improved glycemic control in the first trimester of pregnancy, reduced preterm delivery, and SGA rate reflected positively on the substantial reduction of 54% in perinatal mortality rate in women who attended PCC compared to those who did not (Fig 7, Table 6).

Meta-analysis results showed that the effect of PCC on maternal hypoglycemia was of very low-certainty level. The three included studies had a low risk of bias but a high level of heterogeneity and inconsistency of direction of effect. Studies included were conducted in different time periods, during which time many innovations were made in the management of diabetes, which explains the heterogeneity level (S3 File, Fig 8). Direction, magnitude and certainty about the effect of PCC on maternal hypoglycemia may change with conduction of additional studies addressing this outcome"

We are not surprised that PCC had little or no effects on some outcomes such as macrosomia, shoulder dystocia and CS delivery rate, which may be influenced by perinatal care rather than PCC. Similarly, PCC had no effect on miscarriage rate, this can be explained by the late attendance of the control group for antenatal care by which time many events of miscarriage had already occurred. This assumption is further supported by the significance of early booking for antenatal care of the intervention group shown in this review.

Based on our previous systematic review results, an economic evaluation study found that pregestational diabetes lifetime societal cost is $5.5 billion. However, the study did not evaluate the cost of the recommended universal PCC and the amount of saving with such implementation [13]. Another recently published study [50] showed relatively low saving by provision of PCC to diabetic women compared to routine care. This was explained by the improvement in obstetrics care which may have attenuated potential savings in addition to the poor utilization of PCC as only 40–60% of the target population attend the service [50]. It is worth noting that these studies were conducted in high income countries which makes it difficult to generalize the results to Low and Middle-Income Countries (LMICs) with different economic constrains and health services provision.

In most settings, nearly 50% of pregnancies complicated with diabetes are unplanned, hence this group of women are unlikely to attend PCC service [91–93]. Qualitative studies which investigated the reasons behind poor utilization of PCC showed that women with diabetes who did not attend PCC are more frequently unmarried, have modest education attainment and are unemployed [94]. Other factors include lack of knowledge and attitude of women with diabetes towards fertility, contraception, and the negative message about complications of diabetes in pregnancy rather than the benefits of PCC they tend to receive from

healthcare providers [95]. However, nation-wide programs, which addressed many barriers to the utilization of PCC, had moderate success in increasing attendance for PCC for women with diabetes [50].

It is worth mentioning that all studies included in this review were conducted in high income countries, which may have underestimated the effect of PCC on many outcomes considering the projected increased prevalence of pregestational diabetes in LMICs and the limited resources for antenatal and neonatal care.

## Strengths and limitations

This review is a comprehensive assessment of all important maternal, perinatal outcomes which could be improved by a variety of interventions in the preconception period. The review included a large number of studies and participants. The use of the GRADE tool to evaluate the body of evidence has improved our certainty about the effectiveness of PCC for the main outcomes. The results of this review concur with previously published reviews on the effectiveness of PCC [9, 16]. However, it provides higher quality of evidence with high certainty for the main important outcomes indicating that further research is unlikely to change the conclusion about the effectiveness of PCC in these outcomes.

We are aware of the limitations of this review including the uncertainty about the feasibility and the applicability of PCC in LMICs, as all the included studies were conducted in high income countries, especially if we consider the high cost of such programs. All the studies which contributed data to this review are observational, which downgraded the body of evidence from high to low before even considering other factors which affect the certainty about the direction and size of the effect of intervention on the outcomes. However, it is unlikely to conduct trials examining an intervention such as PCC because it is unethical to randomize diabetic women to receive or not to receive PCC. The only trial included in this review had major biases because it allowed participants to shift between the intervention and the control groups, hence the results lacked internal validity.

**Implication to practice.** New strategies for incorporating PCC in ongoing healthcare services, such as adults' diabetic clinics and primary healthcare may prove to be cost effective and improve the feasibility and applicability of PCC globally.

Incorporation of health education about contraception, fertility of women with diabetes, and the importance of pregnancy planning in the services of diabetic clinics may improve the uptake of PCC.

**Implication to research.** Further research in interventions for improving pregnancy planning and increase utilization of PCC in different communities may improve our understanding of the poor utilization of PCC and suggest areas for improvement. In addition, research to investigate important outcomes, which are still surrounded by uncertainty, such as the association between PCC and maternal hypoglycemia, should be encouraged. There is lack of studies addressing these problems in low-income countries that raise the need for future research both quantitative and qualitative.

## Conclusion

PCC for women with pre-gestational type 1 or type 2 diabetes mellitus is effective in improving rates of congenital malformations. In addition, it may improve the risk of preterm delivery and admission to NICU. PCC probably reduces maternal HbA1C in the first trimester of pregnancy, perinatal mortality and SGA. There is uncertainty regarding the effects of PCC on early booking for antenatal care or maternal hypoglycemia during the first trimester of pregnancy. PCC has little or no effect on other maternal and perinatal outcomes.

## Supporting information

**S1 Checklist. PRISMA 2009 checklist PCC.**
(DOC)

**S1 File. Search strategy.**
(DOCX)

**S2 File. Excluded studies.**
(DOCX)

**S3 File. Sensitivity analysis.**
(DOCX)

**S1 Fig. Forest plots.**
(DOCX)

## Acknowledgments

We would like to thank Roaa Elkouny and A'alaa Abdelrahman for proof editing the final version of the manuscript.

## Author Contributions

**Conceptualization:** Hayfaa A. Wahabi, Amel Fayed, Hala Elmorshedy.

**Data curation:** Hayfaa A. Wahabi, Samia Esmaeil, Hala Elmorshedy, Maher A. Titi, Yasser S. Amer, Rasmieh A. Alzeidan, Abdulaziz A. Alodhayani, Elshazaly Saeed, Khawater H. Bahkali, Melissa K. Kahili-Heede, Amr Jamal.

**Formal analysis:** Hayfaa A. Wahabi, Amel Fayed, Samia Esmaeil, Hala Elmorshedy, Maher A. Titi, Yasser S. Amer, Rasmieh A. Alzeidan, Abdulaziz A. Alodhayani, Elshazaly Saeed, Khawater H. Bahkali, Yasser Sabr.

**Investigation:** Hala Elmorshedy, Elshazaly Saeed.

**Methodology:** Hayfaa A. Wahabi, Amel Fayed, Samia Esmaeil, Hala Elmorshedy, Maher A. Titi, Yasser S. Amer, Rasmieh A. Alzeidan.

**Project administration:** Hayfaa A. Wahabi, Samia Esmaeil.

**Resources:** Melissa K. Kahili-Heede, Amr Jamal.

**Software:** Amel Fayed, Yasser S. Amer.

**Supervision:** Hayfaa A. Wahabi.

**Validation:** Hayfaa A. Wahabi, Hala Elmorshedy.

**Writing – original draft:** Hayfaa A. Wahabi, Amel Fayed, Hala Elmorshedy, Maher A. Titi.

**Writing – review & editing:** Hayfaa A. Wahabi, Amel Fayed, Hala Elmorshedy.

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
