## [Decision Letter · Decision Letter 0]

29 Jan 2020

PONE-D-19-30647

Systematic review and meta-analysis of the effectiveness of pre-pregnancy care for women with diabetes for improving maternal and fetal outcomes

PLOS ONE

Dear Dr. Fayed,

Thank you for submitting your manuscript to PLOS ONE. After careful consideration, we feel that it has merit but does not fully meet PLOS ONE’s publication criteria as it currently stands. Therefore, we invite you to submit a revised version of the manuscript that addresses the points raised during the review process.

Would you choose to submit a revised manuscript, please address all the comments made by the reviewers.

We would appreciate receiving your revised manuscript by Mar 14 2020 11:59PM. To enhance the reproducibility of your results, we recommend that if applicable you deposit your laboratory protocols in protocols.io, where a protocol can be assigned its own identifier (DOI) such that it can be cited independently in the future. For instructions see: http://journals.plos.org/plosone/s/submission-guidelines#loc-laboratory-protocols

We look forward to receiving your revised manuscript.

Kind regards,

Umberto Simeoni

Academic Editor

PLOS ONE

Journal Requirements:

Reviewers' comments:

Reviewer's Responses to Questions

**Comments to the Author**

1. Is the manuscript technically sound, and do the data support the conclusions?

Reviewer #1: Partly

Reviewer #2: Yes

2. Has the statistical analysis been performed appropriately and rigorously? 

Reviewer #1: Yes

Reviewer #2: I Don't Know

3. Have the authors made all data underlying the findings in their manuscript fully available?

Reviewer #1: Yes

Reviewer #2: Yes

4. Is the manuscript presented in an intelligible fashion and written in standard English?

Reviewer #1: No

Reviewer #2: Yes

5. Review Comments to the Author

Reviewer #1: Dr Amel Fayed (corresponding author) and co-workers have conducted a systematic review and meta-analysis of the effect of pre-pregnancy care (pre-conception care, PCC) for women with diabetes on maternal and fetal outcomes. The rationale for the study is that since last review also conducted by the first author (dr Wahabi) new studies have been published and a tool for assessing the quality of evidence of outcomes in studies has come into use, the Grading of Recommendations, Assessment, Development and Evaluation (GRADE).

In addition to and to explain my comments to authors above I have some questions and suggestions to the authors, specifically, on the Result section.

Title.

I suggest that the same words are used in the title as in the manuscript i.e. pre-conception care. Not only complications in fetuses are studied but also complications in the newborn, maybe offspring or infants would be more appropriate words.

Abstract.

I suggest that the results in the abstract should be re-evaluated, please see below my comments on the result in the main text. Could you also give the number of women with type 1 and type 2 diabetes included in the studies?

Introduction.

The authors describe the importance of well-regulated glucose levels in pregnant women with diabetes, especially during the first trimester. Further, they state that management of diabetes is a challenge in low- and middle-income countries. Here the burden of diabetes is higher in the younger population thus increasing the risk of complications during pregnancy and in the fetus and newborn child. The variation of uptake of recommendations in different socioeconomic strata in all countries is also problematic. These are important issues, but the current study does not specifically address them as the review includes studies from high-income countries. The reasons for lack of knowledge of how to manage diabetes in low resource and socioeconomic under-privileged regions and what to do to amend this should be addressed in the Discussion.

Methods.

The description of search methods, study selection and identification, and data extraction are well described. Under subheading Quality Assessment the score using stars should be described. What does the stars stand for in the different domains? Describe this process in more detail. In Table 4 for stars are used for bias assessment and their use should be explained as this assessment is crucial for the results of the meta-analyses. Reference #20 is wrong (line 193), it should be #21. Line 195; the word trials is misleading as the current literature review only comprise one trial, all other studies are observational.

Assessment of the quality of the evidence (page 9). This section should be thoroughly revised. The symbols of the GRADE tool (used in table 6) should be described and how the design of a study gives the study a default grade should be explained. For instance all observational studies are not default of low quality of evidence. Well-performed cohort studies have usually moderate quality of evidence. Further, reasons for down and up-grading the quality should be summarized, preferably in a supplementary table. Up-grading could also be done if there is a dose-response effect not only if the effect of intervention is great. Albeit, upgrading should be used with care.

Some details: line 212 the table Summary of findings in the manuscript should be referred to.

The statistical analysis is performed with statistical well-known tools and seems appropriate.

Results.

The numbers of full text article is not the same in the text (n=75) and in Figure 1 (n=76) the same applies to excluded studies n=35 and n=36 respectively, please check and revise.

Also check the numbers in table 1. Two studies in the table are not mentioned in the text (20 and 31). Line 261: “..16 were retrospective studies..” but the number within brackets is 17. Under subheading Interventions the references in the text are not congruent with those in table 1. The same apply to the references in the text with subheading Outcome measures. In the table two connected studies are presented as one which is confusing. It would be easier for the reader to check the literature if the articles in the table are presented consecutively according to their reference number.

Subheading “Effects of intervention”.

Lines 5-8. Gestational age at booking for antenatal care. The statement on quality of evidence is not supported by the data, albeit the effect of PCC is small. Five studies is not a small number of studies and the number of individuals included is large and the risk of bias as stated by the authors is moderate according to table 6, this should be congruent with the text. The number of studies, if more than one, is not a reason for down-grading of evidence. I suggest your conclusion of very low-quality of evidence should be reconsidered. Figure 2: the green symbol of reference Rosenn 1991 should be explained in the Table text (large drop-out, 50%). Usually studies with a drop-out over 20% are excluded from analysis. Reconsider if this study should be excluded from the analysis? The same applies for Rosenn in Figure 4.

Lines 12-17. Congenital malformation. One study should be excluded from the Forrest plot since it does not contribute to the result (Dunne 1999). Also an analysis excluding studies with high risk for bias should be performed in order to investigate the impact on the outcome, see also Discussion, line 117-121. Please also check the congruence of numbers in text and table.

Line 21-28. HbA1c. The conclusion of high-quality of evidence is not supported by Table 6 where the outcome is designated moderate quality of evidence.

Lines 32-35. Maternal hypoglycemia. All three studies in the meta-analysis are assessed by the authors having low risk of bias. Reconsider the conclusion of very low-quality of evidence. In Figure 5 the text on x-axis (experimental and control) should be replaced to PCC and non-PCC.

Lines 39-42. Preterm delivery. The authors state that the outcome has moderate-quality of evidence due to small effect. The reason for small effect could be that this is actually the true outcome. Small effect size is not conferring down-grading of evidence as is the opposite (large effect could increase the quality of evidence). All included studies (except Dunne 1999) have low risk of bias (the most important weight for assessment of evidence as stated on page 34 lines 25-26 in the manuscript and I fully agree with that). The conclusion should be reconsidered. In the Discussion the global burden of preterm delivery is discussed, the limitation of this review is that the studies are from high-income countries. There is a lack of studies from under-privileged countries where PCC could have a greater impact on this outcome.

Lines 46-50. Perinatal mortality. Similarly, to “preterm delivery”. The quality of evidence is not only affected by the effect size (here presumed to increase it) but relies on the quality of the included studies. The reduced risk of perinatal mortality is mainly driven by two older studies Boulot 2003 (low-risk of bias) and Willhoite 1993 (high risk of bias due to differences in baseline characteristics for instance duration of diabetes and difference in prenatal care in the two groups). I suggest that Willhoite and Dunne 1999 is excluded from analysis because of the high risk of bias for those studies. The data should be re-analyzed.

Lines 56-58. Small for gestational age. The included 6 studies have all (except Dunne 1999) low risk of bias and a large number of participants. Reconsider the conclusion of low quality evidence according to my comments to the preceding analyses of preterm delivery and SGA.

Lines 62-65. NICU admission. Low risk of bias for three studies including a large number of participant. Reconsider the conclusion.

Line 69-75. Other outcomes. Many of these outcomes are based on studies with low risk of bias (only shoulder dystocia has low precision (due to low number of events). Some analyses point towards no effect of PCC on outcome with a high quality of evidence. Reconsider the results.

Publication bias. I don’t understand how heterogeneity in outcome can explain publication bias in reporting of HbA1c. Can you explain this further?

Can you present data on the number of patients with type 1 and type 2 diabetes in the studies? Were there differences in outcome of PCC in the two conditions?

Discussion.

The authors state that PCC has an impact on several outcomes concerning the mother and the off-spring. The impact might be less or even non-existent in other outcomes should also be reported. The evidence for no difference between intervention and no intervention could be of sufficient quality (for instance if large studies were included in the analysis). This might be very valuable knowledge for maternity health care planners.

Line 117. The authors state that the high quality of evidence on PCC and congenital malformation is mainly based on the large effect size. The upgrading of evidence due to effect size should be used with great caution. The high risk of bias in nearly half of the studies should also be a caveat making the meta-analysis more uncertain, see also my comments to this outcome in the Result section. Still, moderate quality of evidence could be good enough for introduction of an intervention and a greater size of effect have a role for this decision.

Line 122-136. One reason for the lower risk of congenital malformation is better control of blood glucose during pregnancy. Other studies (78-81) have shown a dose-response effect of HbA1c level on congenital malformations, did the authors find dose-response effects in the current study? If this was the case an upgrading of the quality of evidence could be considered.

The authors point to a very important issue in studies with a great time span. Practice changes over time (i.e. different criteria for diabetes diagnosis, and introduction of continuous glucose measurements) and this could be a reason for excluding older studies or perform analyses of studies from different time intervals. A more comprehensive care of women in fertile ages and in pregnant women with diabetes could also diminish the positive effect of specific PCC programs. The effect of PCC could also vary in areas with different socioeconomic status and between high, middle and low income countries which also the authors state in lines 196-198. This issues could be further elaborated in the sections of implications for practice and research, lines 231-246 and joined with the text from lines 208-211.

Lines 157-169. I don’t agree that there were a small number of studies supporting the effect of PCC on SGA. See my comments in the Result section.

Line 179-181. Can you explain what you mean with the true effect and how it would be substantially different? In what way? See also my comments to this outcome in the Result section.

The result of this review is rather clear. PCC increases the health of mothers with diabetes and their off-spring in high-income countries. A limitation that leaves gaps of knowledge is that studies from middle and low-income countries were lacking. This is especially important as the authors focus on these issues in the Introduction of the manuscript. The authors describe some studies using surveys and qualitative design to investigate women’s attitudes towards pregnancy planning in Britain and the US and one study of risk factors for SGA in Brazil. The results from these studies might help formulate strategies to reach women with diabetes in childbearing age at risk, in underprivileged areas, and in middle and low-income countries.

References.

The references should be scrutinized for typos and incompleteness. For instance #20, #23, #47, #81.

General remarks.

A linguistic revision of the text is needed and there are many typos and lack of spaces in the text.

Reviewer #2: The large amount of observations makes it a little difficult to follow the statistical procedures. Trying to calculate some figures, I find the total number of participants in the cohort studies to be 8361, in the article given as 8324 (p. 10, l. 262).

Data for some participants are missing. I found data from 8040 participants possible for meta-analysis, in the paper the number is 8026 (p. 34, l. 3).

Similarly, I calculated the number of subjects studied for HbA1c to be 4907, in the paper given as 4927.

I think the authors are right in their calculations. The results will probably be unchanged.

Have I misunderstood some details on pp. 34-37 (effects of intervention)?

Congenital malformation - text: 24 studies, 5856 women, Table 5: 25 studies, 5903 women

Perinatal mortality - text: ten studies, 3071 women, Table 5: 9 studies, 3024 women

A few remarks on writing:

p. 8, l. 189 and l. 190 - should probably read "low risk of bias" and "high risk of bias", respectively, instead of parentheses. Also: - at least one STAR instead of start.

p.8, l. 204 - represents the SLANDERED error ... should probably read STANDARD error?

p. 9, l. 225 - we assessED the quality ...

p. 11, l. 264 - and 180 women in the trial, going through 270 pregnancies (it seems that the calculations involve pregnancies and not women?)

p. 17, regarding study 14/Gunton (45) What does "Total N= of women: 61" mean?

and regarding study 15/Gunton (45) - "Total Number of women:31"?

p. 40, l. 101 - slightly earliER booking

l. 121 - participants were FROM studies ...

p. 41, l. 131 -27% of participants were FROM studies ...

l. 154 - were FROM one study ...

p. 42, l. 168 - participants were FROM one study ...

l. 177 - the participants were FROM two studies ...

p. 43, L. 197 - LMICs should probably be defined, although many readers would know the abbreviation (Low and Middle Income Countries?)

Regarding health economics, the different costs might all be given in the same currency, for comparison.

6. PLOS authors have the option to publish the peer review history of their article (what does this mean?). If published, this will include your full peer review and any attached files.

Reviewer #1: No

Reviewer #2: No

---

## [Author Response · Author response to Decision Letter 0]

10 Mar 2020

Dear Editors,

We would like to thank the reviewers for their informative and constructive comments, based on their directions, we considered their comments and revised all reported assessments numbers and figures.

Kindly find below our point-by-point reply for their comments:

Reviewer 1

We thank the reviewer for his/her meticulous revision of all the outcomes, in some instances we agreed with some of his/her comments and changed our assessment of the GRADE of evidence on certain outcomes. However, we need to explain that the GRADE tool is mainly used to give the policy makers and those who implement evidence some direction about the expected degree of success in improving the particular outcome if the intervention is implemented. Hence, the importance of the effect size, heterogeneity, CI crossing no effect line and the consistency of the direction of outcomes in all the studies and not only the methodological quality of evidence including bias. Another important consideration in our assessment is that all observational studies, based on the GRADE approach (not our decision), start at a score of (low), which is the third in a four-point scale assessing our certainty on the evidence 

Reviewer 1 comments Action/comments 

Title.

I suggest that the same words are used in the title as in the manuscript i.e. pre-conception care. Not only complications in fetuses are studied but also complications in the newborn, maybe offspring or infants would be more appropriate words. Reply: According to WHO, the perinatal period commences at 22 completed weeks (154 days) of gestation and ends seven completed days after birth which includes all the events reported in the current study. So we changed the term “fetal” to Perinatal” in the title. 

Abstract.

I suggest that the results in the abstract should be re-evaluated, please see below my comments on the result in the main text. Could you also give the number of women with type 1 and type 2 diabetes included in the studies?

 Reply: Some of the results have been modified according to the reviewer’s comments (please see reply to reviewer in results section). We could not find the exact number of women with type 1 and type II diabetes because some studies didn’t specify the numbers in each group, most probably because the effect of PCC does change with the type of diabetes unless there were vascular complications related to diabetes and those we have considered when we assessed the presence of confounders in our assessment of bias in the included studies (please see table 4).

.

The authors describe the importance of well-regulated glucose levels in pregnant women with diabetes, especially during the first trimester. Further, they state that management of diabetes is a challenge in low- and middle-income countries. Here the burden of diabetes is higher in the younger population thus increasing the risk of complications during pregnancy and in the fetus and newborn child. The variation of uptake of recommendations in different socioeconomic strata in all countries is also problematic. These are important issues, but the current study does not specifically address them as the review includes studies from high-income countries. The reasons for lack of knowledge of how to manage diabetes in low resource and socioeconomic under-privileged regions and what to do to amend this should be addressed in the discussion.

 Reply: We included this information in the in introduction to portray a complete picture about the epidemiology od diabetes is a common health problem all over the world especially young age group and in different countries, hence, the importance of this review to introduce PCC as a preventive measure to most of the complications of diabetes in pregnancy. Further details were included in the discussion

Methods:

The description of search methods, study selection and identification, and data extraction are well described. Under subheading Quality Assessment the score using stars should be described. What does the stars stand for in the different domains? Describe this process in more detail.

 Reply: the details of the tool (The Newcastle-Ottawa Scale), has been explained in details in the methods section with its cited reference. (page 8, lines 186-193)

In Table 4 for stars are used for bias assessment and their use should be explained as this assessment is crucial for the results of the meta-analyses. Reply: the details of staring studies was explained in details in the methods section with its cited reference. (page 8, lines 186-193)

Reference #20 is wrong (line 193), it should be #21. Corrected

Line 195; the word trials is misleading as the current literature review only comprise one trial, all other studies are observational.

 Corrected to studies

Assessment of the quality of the evidence (page 9). This section should be thoroughly revised. The symbols of the GRADE tool (used in table 6) should be described and how the design of a study gives the study a default grade should be explained. For instance, all observational studies are not default of low quality of evidence. Well-performed cohort studies have usually moderate quality of evidence. Further, reasons for down and up-grading the quality should be summarized, preferably in a supplementary table. Up-grading could also be done if there is a dose-response effect not only if the effect of intervention is great. Albeit, upgrading should be used with care.

Some details: line 212 the table Summary of findings in the manuscript should be referred to. The statistical analysis is performed with statistical well-known tools and seems appropriate. Reply: All the details of the GRADE approach are addressed clearly in the methods section according to reference 21. 

Kindly find this reference for more explanation and clarifications of the GRADE approach. https://www.jclinepi.com/article/S0895-4356(13)00057-7/fulltext 

Please refer to the explanations at the top of the reply to reviewers

The numbers of full text article is not the same in the text (n=75) and in Figure 1 (n=76) the same applies to excluded studies n=35 and n=36 respectively, please check and revise. Revised and corrected

Also check the numbers in table 1. Two studies in the table are not mentioned in the text (20 and 31).

 All studies mentioned in the table and text were revised and matched for all outcomes

Line 261: “..16 were retrospective studies..” but the number within brackets is 17 All studies mentioned in the table and text were revised and matched for all outcomes

Under subheading Interventions the references in the text are not congruent with those in table 1 All studies mentioned in the table and text were revised and matched for all outcomes

The same apply to the references in the text with subheading Outcome measures. All studies mentioned in the table and text were revised and matched for all outcomes

In the table two connected studies are presented as one which is confusing. It would be easier for the reader to check the literature if the articles in the table are presented consecutively according to their reference number. Unfortunately, this is nor possible because some authors (e,g. Temple ) has more than one publication in two different medical journals but are different outcomes for the same cohort. For the reader to refer to the certain study they have to find the correct citation. 

Subheading “Effects of intervention”.

Lines 5-8. Gestational age at booking for antenatal care. The statement on quality of evidence is not supported by the data, albeit the effect of PCC is small. Five studies are not a small number of studies and the number of individuals included is large and the risk of bias as stated by the authors is moderate according to table 6, this should be congruent with the text. The number of studies, if more than one, is not a reason for down-grading of evidence. I suggest your conclusion of very low-quality of evidence should be reconsidered. Reply: The quality of evidence was downgraded from low-grade (observational study) to very low -grade due to high- risk of bias in the study with the largest weight (Rossen 1991) and high unexplained heterogeneity (table 6).

Figure 2: the green symbol of reference Rosenn 1991 should be explained in the Table text (large drop-out, 50%). Usually studies with a drop-out over 20% are excluded from analysis. Reconsider if this study should be excluded from the analysis? The same applies for Rosenn in Figure 4. Reply: In the protocol, we did not define drop-off of more than 20% to indicate study exclusion. Loss of follow up is appraised in the NOS for assessment of bias. 

The green symbol is present in all the studies in this forest plot and it indicates the roughly the (mean). It is more prominent for the Rosenn 1999 because the small confidence interval (-1.39 to -1.21) which masked the arms of the confidence interval (the two ends of the black line).

Lines 12-17. Congenital malformation. One study should be excluded from the Forrest plot since it does not contribute to the result (Dunne 1999). Reply: Dunne 1999 reported the congenital anomalies and they reported no cases of congenital anomalies in the intervention or the control groups as (ZERO) is a number and the number of participants should be included in the denominator of the pooled estimate we included the study in the forest Plot.

Kindly check this reference:

https://bmcmedresmethodol.biomedcentral.com/articles/10.1186/1471-2288-7-5

Also an analysis excluding studies with high risk for bias should be performed in order to investigate the impact on the outcome, see also Discussion, line 117-121. Please also check the congruence of numbers in text and table.

 Reply: According to our protocol, we did not plan to do sensitivity analysis based on risk of bias as it is considered in the GRADE approach for quality of evidence. Additionally, kindly see below one example of the forest plot of HbA1c where we excluded all high biased studies (10 studies) however, the pooled effect of PCC on the reduction of HbA1c stayed nearly the same.

Line 21-28. HbA1c. The conclusion of high-quality of evidence is not supported by Table 6 where the outcome is designated moderate quality of evidence.

 Reply: we considered the number of participants in studies with high risk of bias versus number in studies with low risk. We added the number of participants in explaining GRADE evaluation in each outcome.

Lines 32-35. Maternal hypoglycemia. All three studies in the meta-analysis are assessed by the authors having low risk of bias. Reconsider the conclusion of very low-quality of evidence. Reply: Please consider that we start from low certainty because of observational studies. From the forest plot, there is no consistency (different direction of effect) , large CI (1.07-1.79) which is not precise and high heterogeneity all these factors affected our certainty of evidence to be very low from low. It means that we are uncertain if the PCC if implemented mothers will have hypoglycemia or not !!!

In Figure 5 the text on x-axis (experimental and control) should be replaced to PCC and non-PCC. Reply: Corrected

Lines 39-42. Preterm delivery. The authors state that the outcome has moderate-quality of evidence due to small effect. The reason for small effect could be that this is actually the true outcome. Small effect size is not conferring down-grading of evidence as is the opposite (large effect could increase the quality of evidence). All included studies (except Dunne 1999) have low risk of bias (the most important weight for assessment of evidence as stated on page 34 lines 25-26 in the manuscript and I fully agree with that). The conclusion should be reconsidered. Reply: We have actually upgraded the evidence from low to moderate due to the reasons you have mentioned in your comments, however we could not improve our certainty of the effect more because of the small effect size. The GRADE approach considers the implications to clinical practice which can be extracted from the body of evidence, e.g does the evidence support that if PCC is implemented one should expect that preterm delivery will be reduced? If the effect size is small then the certainty that will definitely result in reduction of pre-term delivery will be lower because other factors (e.g maternal age, parity, socioeconomic factors…) may play a bigger role in reduction of PTD. Hence the moderate quality of evidence 

Please see up/downgrading of the quality of evidence is according to guidelines of GRADE.

https://www.jclinepi.com/article/S0895-4356(13)00057-7/fulltext 

In the Discussion the global burden of preterm delivery is discussed, the limitation of this review is that the studies are from high-income countries. There is a lack of studies from under-privileged countries where PCC could have a greater impact on this outcome. 

 We have suggested more than one solution for the provision of PCC globally. Please refer to pp 41-42 lines 215-220 

Lines 46-50. Perinatal mortality. Similarly, to “preterm delivery”. The quality of evidence is not only affected by the effect size (here presumed to increase it) but relies on the quality of the included studies. The reduced risk of perinatal mortality is mainly driven by two older studies Boulot 2003 (low-risk of bias) and Willhoite 1993 (high risk of bias due to differences in baseline characteristics for instance duration of diabetes and difference in prenatal care in the two groups). I suggest that Willhoite and Dunne 1999 is excluded from analysis because of the high risk of bias for those studies. The data should be re-analyzed. Reply: According to our protocol which was published in the, we did not plan to do sensitivity analysis based on risk of bias, based on the fact that risk of bias of the whole body of evidence (all nine studies) is one of five factors considered in the GRADE approach for quality of evidence. Please refer to the above replies for assessment of GRADE OF EVIDENCE 

(Downgrading of the quality of evidence is according to guidelines of GRADE.

https://www.jclinepi.com/article/S0895-4356(13)00057-7/fulltext 

Lines 56-58. Small for gestational age. The included 6 studies have all (except Dunne 1999) low risk of bias and a large number of participants. Reconsider the conclusion of low quality evidence according to my comments to the preceding analyses of preterm delivery and SGA.

Lines 62-65. NICU admission. Low risk of bias for three studies including a large number of participant. Reconsider the conclusion. Reply: Agree, reconsidered please see text

Line 69-75. Other outcomes. Many of these outcomes are based on studies with low risk of bias (only shoulder dystocia has low precision (due to low number of events). Some analyses point towards no effect of PCC on outcome with a high quality of evidence. Reconsider the results.

 Reply: Please look at the CIs of all these outcomes which crossed the no effect line (which is 1), which means it may be effective and it may not be effective (irrespective of all other factors of upgrading or down grading of evidence). In addition, all observational studies started from (low) GRADE and the reasons for upgrading have been mentioned in our methodology in p 9, lines 218-223 

Publication bias. I don’t understand how heterogeneity in outcome can explain publication bias in reporting of HbA1c. Can you explain this further? Reply: Heterogeneity does not give publication bias but when there is marked heterogeneity the funnel plot gives a distribution of the studies similar to publication bias. 

Kindly refer to reference 69 Terrine 2005 for further explanation

Can you present data on the number of patients with type 1 and type 2 diabetes in the studies? Were there differences in outcome of PCC in the two conditions? type 1 and type II patients were reported collectively as some studies did not report the outcomes of each group separately.

Discussion.

The authors state that PCC has an impact on several outcomes concerning the mother and the off-spring. The impact might be less or even non-existent in other outcomes should also be reported. The evidence for no difference between intervention and no intervention could be of sufficient quality (for instance if large studies were included in the analysis). This might be very valuable knowledge for maternity health care planners. Reply: All the outcomes were summarized in the first paragraph and then most of them were elaborated furthermore in the discussion. Other outcomes were discussed on P 40 lines 167-172

Line 117. The authors state that the high quality of evidence on PCC and congenital malformation is mainly based on the large effect size. The upgrading of evidence due to effect size should be used with great caution. The high risk of bias in nearly half of the studies should also be a caveat making the meta-analysis more uncertain, see also my comments to this outcome in the Result section. Still, moderate quality of evidence could be good enough for introduction of an intervention and a greater size of effect have a role for this decision.

 Reply: revised and explained please refer to the text

Line 122-136. One reason for the lower risk of congenital malformation is better control of blood glucose during pregnancy. Other studies (78-81) have shown a dose-response effect of HbA1c level on congenital malformations, did the authors find dose-response effects in the current study? If this was the case an upgrading of the quality of evidence could be considered.

 Reply: these data are not available in the included study to perform this analysis.

The authors point to a very important issue in studies with a great time span. Practice changes over time (i.e. different criteria for diabetes diagnosis, and introduction of continuous glucose measurements) and this could be a reason for excluding older studies or perform analyses of studies from different time intervals. A more comprehensive care of women in fertile ages and in pregnant women with diabetes could also diminish the positive effect of specific PCC programs. The effect of PCC could also vary in areas with different socioeconomic status and between high, middle and low income countries which also the authors state in lines 196-198. This issues could be further elaborated in the sections of implications for practice and research, lines 231-246 and joined with the text from lines 208-211. Reply: in the protocol, sensitivity analysis was planned for unexplained high heterogeneity only, not for bias or temporal reasons

Lines 157-169. I don’t agree that there were a small number of studies supporting the effect of PCC on SGA. See my comments in the Result section. Reply: Corrected

Line 179-181. Can you explain what you mean with the true effect and how it would be substantially different? In what way? See also my comments to this outcome in the Result section. Reply: As a sequence of the uncertainty of the available evidence, the true effect can be different from the reported one either in direction or magnitude or both.

The result of this review is rather clear. PCC increases the health of mothers with diabetes and their off-spring in high-income countries. A limitation that leaves gaps of knowledge is that studies from middle and low-income countries were lacking. This is especially important as the authors focus on these issues in the Introduction of the manuscript. 

The authors describe some studies using surveys and qualitative design to investigate women’s attitudes towards pregnancy planning in Britain and the US and one study of risk factors for SGA in Brazil. The results from these studies might help formulate strategies to reach women with diabetes in childbearing age at risk, in underprivileged areas, and in middle and low-income countries.

 We had to discuss this issue in the discussion and not in the introduction because it became apparent after we have done the review 

Indeed these studies will be helpful especially if they collectively analyzed in a qualitative review. However we cannot comment on that because we don’t know how many studies are available and what their quality are 

References.

The references should be scrutinized for typos and incompleteness. For instance, #20, #23, #47, #81. Reply: corrected

General remarks.

A linguistic revision of the text is needed and there are many typos and lack of spaces in the text. Reply: Corrected

 

Reviewer 2 

We thank the reviewer for his/her meticulous review of the number included in the review which was of great help to us 

Reviewer 2 comments Action/comments 

Trying to calculate some figures, I find the total number of participants in the cohort studies to be 8361, in the article given as 8324 (p. 10, l. 262). Recalculation: PCC:3213, NOPCC: 4986. Total: 8199

• In Gunton(2000): reported a total of 61 women.

24 pregnancies were reported in PCC (some participants had more than one pregnancy) and 69 pregnancies were reported in NO-PCC (some participants had more than one pregnancy)

• Gunton (2002): reported a total of 31 women. 19 pregnancies were reported in PCC (some participants had more than one pregnancy) and 16 pregnancies were reported (some participants had more than one pregnancy

• Kallas-Koeman (2012): Data available for only 71and 150 participants.

Data for some participants are missing. I found data from 8040 participants possible for meta-analysis, in the paper the number is 8026 (p. 34, l. 3). Corrected:

8199 participants 

I calculated the number of subjects studied for HbA1c to be 4907, in the paper given as 4927. Recalculated: the correct number is 4927, please see first forest plots which is matching the number in table 1 and in the text

Have I misunderstood some details on pp. 34-37 (effects of intervention)?

 Please note that some forest plots are shown in supplementary files (only 8 in the manuscript), all outcomes are listed in the text. Summary of findings table includes only the important outcomes.

Congenital malformations - text: 24 studies, 5856 women, Table 5: 25 studies, 5903 women Revised and Corrected:

: the total number of included studies is 25 with a total of 5903 participants. The number corrected in the text. 

Perinatal mortality - text: ten studies, 3071 women, Table 5: 9 studies, 3024 women Revised and corrected:

10 studies with 3071 women 

p. 8, l. 189 and l. 190 - should probably read "low risk of bias" and "high risk of bias", respectively, instead of parentheses. Also: - at least one STAR instead of start. Revised and corrected:

p.8, l. 204 - represents the SLANDERED error ... should probably read STANDARD error? Revised and corrected:

p. 9, l. 225 - we assessED the quality ... Revised and corrected:

p. 11, l. 264 - and 180 women in the trial, going through 270 pregnancies (it seems that the calculations involve pregnancies and not women?) Revised and corrected:

p. 17, regarding study 14/Gunton (45) What does "Total N= of women: 61" mean? Revised and corrected:

• The numbers rewritten and explained in table 1 as following 

• In Gunton(2000): 

24 pregnancies were reported in PCC (some participants had more than one pregnancy) and 69 pregnancies were reported in NO PCC (some participants had more than one pregnancy)

and regarding study 15/Gunton (45) - "Total Number of women:31"? Revised and corrected:

• The numbers rewritten and explained in table 1 as following

• Gunton (2002): reported as Total N= of women: 31 in table 1. 19 pregnancies were reported in PCC (some participants had more than one pregnancy) and 16 pregnancies were reported (some participants had more than one pregnancy

p. 40, l. 101 - slightly earliER booking Revised and corrected:

l. 121 - participants were FROM studies ... Revised and corrected:

p. 41, l. 131 -27% of participants were FROM studies ... Revised and corrected:

l. 154 - were FROM one study ... Revised and corrected:

p. 42, l. 168 - participants were FROM one study ... Revised and corrected:

l. 177 - the participants were FROM two studies ... Revised and corrected:

p. 43, L. 197 - LMICs should probably be defined, although many readers would know the abbreviation (Low and Middle Income Countries?)

Regarding health economics, the different costs might all be given in the same currency, for comparison. Revised and corrected:

---

## [Decision Letter · Decision Letter 1]

26 May 2020

PONE-D-19-30647R1

Systematic review and meta-analysis of the effectiveness of pre-pregnancy care for women with diabetes for improving maternal and Perinatal outcomes

PLOS ONE

Dear Dr. Fayed,

Thank you for submitting your manuscript to PLOS ONE. After careful consideration, we feel that the revised manuscript addresses some of the criticisms made by te reviewers, but still major methodological issues are considered unsatisfactory by the reviewers, including the statistical reviewer we have invited. 

The high risk of bias resulting from the selection of studies is still insufficiently assessed and taken into account in the manuscript.

We would be happy to reconsider a final, revised version of the manuscript, could these major drawbacks be fixed by the authors. However, we would also perfectly understand if you chose not to resubmit, being aware of the work in depth needed to meet the criteria of the journal for publication.

Please note that, still, wording and English language errors are persisting in the manuscript.

We look forward to receiving your revised manuscript.

Kind regards,

Umberto Simeoni

Academic Editor

PLOS ONE

Reviewers' comments:

Reviewer's Responses to Questions

**Comments to the Author**

1. If the authors have adequately addressed your comments raised in a previous round of review and you feel that this manuscript is now acceptable for publication, you may indicate that here to bypass the “Comments to the Author” section, enter your conflict of interest statement in the “Confidential to Editor” section, and submit your "Accept" recommendation.

Reviewer #1: (No Response)

Reviewer #2: (No Response)

Reviewer #3: (No Response)

2. Is the manuscript technically sound, and do the data support the conclusions?

Reviewer #1: Partly

Reviewer #2: Yes

Reviewer #3: (No Response)

3. Has the statistical analysis been performed appropriately and rigorously? 

Reviewer #1: Yes

Reviewer #2: Yes

Reviewer #3: (No Response)

4. Have the authors made all data underlying the findings in their manuscript fully available?

Reviewer #1: Yes

Reviewer #2: Yes

Reviewer #3: (No Response)

5. Is the manuscript presented in an intelligible fashion and written in standard English?

Reviewer #1: No

Reviewer #2: Yes

Reviewer #3: (No Response)

6. Review Comments to the Author

Reviewer #1: Thank you very much for your appreciation of my work with your manuscript.

Dr Amel Fayed (corresponding author) and co-workers have answered to my review and done a lot of changes that are satisfying and thus increased the clarity of the manuscript. Still there are some issues that need to be addressed.

General.

I don’t think that we disagree on the great importance that evidence or lack of evidence for effect or lack of effect of different interventions are studied using systematic reviews and meta-analyses. It is crucial to guide health care staff as well other stakeholder in decisions of interventions demanding much resources. Before GRADING the evidence of different outcomes the included studied are assessed for relevance and quality as you have described. In a conservative assessment, which I propose, studies with high drop-out rate (>30%) and low quality are excluded as the results of these studies are not possible to evaluate. They could be included but then the effect on the outcome must be assessed as the authors have done in presenting the percentage of studies of high risk of bias in the analysis. Observational studies can have very low GRADE score, low GRADE score but some high quality cohort studies can have moderate GRADE score. Thus, the GRADE score does not only depend on the design of the studies but also of their inherent quality. This is very important to recognize in issues that can´t be studied with randomized trials for ethical reasons as in the current systematic review.

Title.

Perinatal is an appropriate word, with lower case p. The word fetal is still used in the manuscript and should be changed for instance in line 135, 314. There might be more, so please scrutinize the manuscript.

Abstract.

Conclusion: I suggest you use the term pre-gestational diabetes as there was no possibility to present the frequencies of typ 1 and type 2, respectively. The assessment of risk of hypoglycemia during first trimester was based on studies of type 1 diabetes, which ought to be pointed out in the manuscript. This lack of information in the rest of literature should be addressed in the manuscript for instance in the section on page 10 under the heading: Participants (line 266).

I can´t see any description of confounders such as other cardiovascular conditions (hypertension, lipid disturbances and obesity) in Table 4 as referred to by the authors.

Introduction.

The authors describe the variation of uptake of recommendations in different socioeconomic strata. In case “Low resource countries” are equivalent to “low income countries” use the latter term throughout the manuscript. If not define what you mean by low resource country. These are important issues the in the Discussion the authors should elaborate what to do to amend this. As there is a lack of studies addressing these problems it have implications for future research both quantitative and qualitative and this should be pointed out under subheading “Implications to research” (line 251).

Methods.

Under subheading Quality Assessment the score using stars is still not described neither in the text nor in the text accompanying Table 4. Further, assessment of the quality of the evidence and the symbols of the GRADE tool should be described, preferably in a supplementary table. I don’t think it is enough to refer to other publications.

Results.

The two connected studies (e.g. Temple) presented as one as they describe different outcomes in the same populations ought to be presented separately as the result are different outcomes if the authors want to make it easier for the reader to check the literature. The same applies if the articles in the table are presented consecutively according to their reference number.

Subheading “Effects of intervention”.

Gestational age at booking for antenatal care. I am satisfied that you omitted the statement of small number of studies. All symbol or acronyms used in a table (or a figure) such as the green dot should be explained in the explaining text or legend, in this case: green dot, calculated mean.

Congenital malformation, I accept your explanation. Line 21 change “form” to “from”.

Line 28 change “form” to “from”.

Maternal hypoglycemia. All three studies in the meta-analysis are assessed by the authors as having low risk of bias which means that they are of high quality although they have low numbers of events. This could increase the risk of bias. More importantly, the oldest (Steel et al) was published 1990 with data from the nineteen seventies and eighties before the meticulous surveillance of b-glucose and new recombinant, ultra-rapid acting insulins were introduced in contrast to Holmes et al. published 2017 and Temple et al who used data from 1990 to 2002. The heterogeneity could thus be caused by secular trends in treatment and surveillance. The 2 modern studies (Holmes and Temple) show no significant difference between PCC and controls. This implicates that maternal hypoglycemia in modern management of type 1 diabetic women (at least in high income countries) is not influenced by PCC. If the authors still consider Holmes and Temple having low risk of bias, I think that the evidence for this outcome should be low-quality, not very-low quality. I would say that you are not totally uncertain if PPC confers hypoglycemia or not compared to non-PCC, but it might not matter, still more studies could alter the evidence.

I think your explanation of the result concerning HbA1c applies here. The explanation of few studies shall be omitted from the text. I presume that your statement of true effect means that using modern data and more research would show if the risk of hypoglycemia is increased or decreased using PCC in a time with continuous glucose measuring (CGM) for an increasing part of patients with type 1 diabetes. You have already mentioned this in implications for research.

Preterm delivery. Omit the sentence “The grade of evidence is considered moderate due to the small effect”. A small effect could be the true outcome. Adjust sentence in lines 52-53 …selective reporting increase our confidence in the outcome of a small reduction (4%) in preterm delivery.

Perinatal mortality. The upgrading is a too high. A reduction of RR to < 0.5 as in the current meta-analysis increase the grade by one star to moderate quality of evidence. RR < 0.2, which is not the case here, gives rise to two stars. The quality of evidence should thus be moderate.

Small for gestational age. The RR is 0.52 not reaching the level <0.50 for upgrading one star. Thus up-grading is not an obvious action. If most included studies have low risk of bias the default grade score could be moderate quality. Upgrading to high is not appropriate here.

Other outcomes. Many of these outcomes are based on studies with low risk of bias. As the authors point out in the reply the CI of most studies cross the line depicting no significant difference between the PCC and non-PCC groups, which means that PCC might have no effect on these outcomes. Many studies with low bias and no difference between groups could render an upgrading, but I can accept your statement of low-certainty evidence.

Discussion.

The impact of PCC might be less or even non-existent in some outcomes. This should be summarized in the first paragraph as it is valuable knowledge for maternity health care planners. The risk for hypoglycemia should be elaborated further as one of the included studies describes management no longer used in high income countries.

Lines 142- 143. Alter to … “moderate quality albeit the effect was size was small”.

The authors point to a very important issue in studies with a great time span. Practice changes over time (i.e. different criteria for diabetes diagnosis, and introduction of continuous glucose measurements) this is especially important for the risk of hypoglycemia during pregnancy, which was much discussed earlier when stricter management began. I persists in the suggestion that you address this, even if you do no sensitivity analysis as this an important outcome that might need further studies with modern treatment. This should be addressed in connection to the last paragraph on page 40, lines 193-195. See also my comments in the result section.

The result of this review is rather clear. PCC increases the health of mothers with diabetes and perinatal period of the off-spring in high-income countries. The authors describe some studies using surveys and qualitative design to investigate women’s attitudes towards pregnancy planning in Britain and the US and one study of risk factors for SGA in Brazil. The results from these studies might help formulate strategies to reach women with diabetes in childbearing age at risk, in underprivileged areas, and in low-income countries. This could be addressed in the section implications for research.

General remarks.

A further linguistic revision of the text is needed and there are still typos in the text.

Reviewer #2: With the large amount of data, the authors should check carefully the numbers in text and tables.

Abstract

The effect on maternal hypoglycemia is reported to be RR 1,42;95% CI: 0,72-2,82 (p 2, l 60) and also in the Effects of intervention (p 32, l 36), while in Tables 5 and 6 it is said to be RR 1,38; 95% CI 1,07-1,79.

Search methods

..all the literature published up to March 2019 ..

This sentence might have included "between 1983 and .." - this information is given in Study Characteristics (p 10, l 268)., One study dated 1982 is also included, maybe this one was found linked to another study?

Table 1

Cohort study 12 (Garcia-Patterson): a miscarriage rate of 13/66 gives 53 continuing pregnancies. The rate of SGA should probably read 1/53.

Similarly, the rate of RDS should probably read 12/119.

Table 2

Macrosomia rate in the NO-PCC group should probably read 4/12.

Assessment of the methodological quality of the included studies

Ref (40) is named Rosen et al (p 26, l 327). The correct spelling is Rosenn, as given in References, Tables 1 and 4.

Effects of intervention

Here the same reference is named Rossen (p 31, l 9).

Gestational age at booking for antenatal care

MD 1,31(probably weeks? as it is translated into approximately ten days) - p 31, l 7.

Discussion

Incremental increase ... (p 38, l 134) is superfluous.

FORM should be corrected to FROM at several places (p 31, l 21, p 32, l 28 and 51, p 33, l 73)

Reviewer #3: Conduct a meta-analysis to evaluate the effectiveness and safety of pre-conception care in improving maternal and perinatal outcomes and evaluate the grade of the body of evidence for each outcome. They identify 36 studies and the meta-analysis results showed that PCC results in large reduction in congenital malformations, lowers HaA1c in the first trimester of pregnancy, lowers the preterm delivery rate

1. Abstract: “the result… that PCC results in large reduction in congenital malformations, …owers HbA1c….” the causal effect was implied in the results. However, the study includes both trial and observational studies. The causal effect should be avoided throughout the manuscript.

2. Line 311. 21 studies were assigned to be at low risk of bias while 15 studies at high risk of bias. A sensitive analysis may be warranted to evaluate the robustness of the findings.

3. Figures 2 and 9. What’s the green rectangle in the figure 2 or blue rectangle in figure 9? Explanation to the figures are needed.

7. PLOS authors have the option to publish the peer review history of their article (what does this mean?). If published, this will include your full peer review and any attached files.

Reviewer #1: No

Reviewer #2: No

Reviewer #3: No

---

## [Author Response · Author response to Decision Letter 1]

19 Jun 2020

Dear Editor and reviewers,

We would like to thank our reviewers for their constructive comments. We agreed with our reviewers on their feedback and all required changes were done accordingly.

Point-by-point reply:

Reviewer #1: Thank you very much for your appreciation of my work with your manuscript.

Dr Amel Fayed (corresponding author) and co-workers have answered to my review and done a lot of changes that are satisfying and thus increased the clarity of the manuscript. Still there are some issues that need to be addressed.

General.

I don’t think that we disagree on the great importance that evidence or lack of evidence for effect or lack of effect of different interventions are studied using systematic reviews and meta-analyses. It is crucial to guide health care staff as well other stakeholder in decisions of interventions demanding much resources. Before GRADING the evidence of different outcomes, the included studied are assessed for relevance and quality as you have described. In a conservative assessment, which I propose, studies with high drop-out rate (>30%) and low quality are excluded as the results of these studies are not possible to evaluate. They could be included but then the effect on the outcome must be assessed as the authors have done in presenting the percentage of studies of high risk of bias in the analysis. Observational studies can have very low-grade score, low GRADE score but some high-quality cohort studies can have moderate GRADE score. Thus, the GRADE score does not only depend on the design of the studies but also of their inherent quality. This is very important to recognize in issues that can´t be studied with randomized trials for ethical reasons as in the current systematic review.

Reply: We totally agree with the reviewer and we have conducted sensitivity analysis excluding high risk of bias studies from the analysis to confirm the statement. Please see page (35,36, supplement 4) 

Title.

• Perinatal is an appropriate word, with lower case p. The word fetal is still used in the manuscript and should be changed for instance in line 135, 314. There might be more, so please scrutinize the manuscript.

Reply: Agree and Done

Abstract.

Conclusion: I suggest you use the term pre-gestational diabetes as there was no possibility to present the frequencies of typ 1 and type 2, respectively. The assessment of risk of hypoglycemia during first trimester was based on studies of type 1 diabetes, which ought to be pointed out in the manuscript. This lack of information in the rest of literature should be addressed in the manuscript for instance in the section on page 10 under the heading: Participants (line 266).

I can´t see any description of confounders such as other cardiovascular conditions (hypertension, lipid disturbances and obesity) in Table 4 as referred to by the authors.

Reply: Agree done as below

• Pregestational diabetes was used, 

• (all reported results among type 1 diabetes) was added in the assessment of maternal hypoglycemia (page 12, line 310-311)

• “Most of studies did not report the differences in the outcomes among type 1 versus type 2 diabetes, as a result, we could not conduct the analysis separately for each type of diabetes” added to participants section, page 11, lines 285,286 

Introduction.

The authors describe the variation of uptake of recommendations in different socioeconomic strata. In case “Low resource countries” are equivalent to “low income countries” use the latter term throughout the manuscript. If not define what you mean by low resource country. These are important issues the in the Discussion the authors should elaborate what to do to amend this. As there is a lack of studies addressing these problems it have implications for future research both quantitative and qualitative and this should be pointed out under subheading “Implications to research” (line 251).

Reply: Reply: Agree done as below

• “Low income countries” was used instead of low resource countries.

• There is lack of studies addressing these problems in the low-income countries that raise the need for future research both quantitative and qualitative, added to implication to research section, page 45, lines 244-246

Methods.

Under subheading Quality Assessment the score using stars is still not described neither in the text nor in the text accompanying Table 4. Further, assessment of the quality of the evidence and the symbols of the GRADE tool should be described, preferably in a supplementary table. I don’t think it is enough to refer to other publications.

Reply: Reply: Agree done as below

• This paragraph was added to the methods section (page 8, lines 224-250)

“The criteria assessed were: participants’ selection, comparability of groups and assessment of outcome for cohort studies. Participants’ selection, comparability of groups, and exposure criteria were used to assess the case-control studies. The maximum number of stars is nine: four stars awarded for selection-selection of exposed and non- exposed, ascertainment of exposure and temporal relation between exposure and outcome-, two for comparability if analysis controlled for confounder- and three stars awarded for outcome if the length of follow up was adequate, no attrition bias, and outcome was assessed independent of exposure Studies at “high risk of bias” score less than six stars or scores no stars in comparability domain irrespective of the number of stars scored.”

• This caption was added to table 4:

“Risk of bias was assessed using the Newcastle-Ottawa Scale (NOS). The number of stars represents the risk of bias, the maximum number of stars is nine, studies were classified as “low risk of bias” if they received a score of six stars or more and there is at least one star in the comparability domain. Studies at “high risk of bias” score less than six stars or scores no stars in comparability domain irrespective of the number of stars scored”

• This caption was added to the GRADE table

a Upgraded to high because of large effect size, consistency of direction of effect, no indirectness of evidence, and no heterogeneity or publication bias. 

b Upgraded to moderate due to the narrow confidence intervals, consistency of direction of effect, no indirectness of evidence, and low risk of bias, no heterogeneity or publication bias.

c Downgraded to very low-grade due to the high risk of bias in the study with the largest weight (Rosenn 1991) and high unexplained heterogeneity

d Upgraded to moderate-certainty level because of low bias (77% of the participants were from studies at low risk of bias), while heterogeneity can be explained by long span of time between the first and the last study (1982 and 2017), The publication bias can be explained with the heterogeneity.

e Downgraded to very low-level certainty because of small number of studies, low bias, explained heterogeneity

f Upgraded to moderate-certainty level because of narrow confidence intervals, consistency of direction of effect, no indirectness of evidence, low risk of bias, low heterogeneity, no evidence of selective reporting.

G Upgraded to moderate-certainty level because the large effect size with precise narrow confidence interval, consistency of direction of effect, no indirectness of evidence, and no heterogeneity and no evidence of selective reporting.

Results.

The two connected studies (e.g. Temple) presented as one as they describe different outcomes in the same populations ought to be presented separately as the result are different outcomes if the authors want to make it easier for the reader to check the literature. The same applies if the articles in the table are presented consecutively according to their reference number.

Reply: Done

 References are cited in order at first mention in the table of included studies, studies are ordered in an alphabetical order to make it easier to list. Results are reported from different studies separately as in (Temple a,b) cited as two different references.

Subheading “Effects of intervention”. However, it is not possible to reference them as one article as they were published in separate journals.

Gestational age at booking for antenatal care. I am satisfied that you omitted the statement of small number of studies. All symbol or acronyms used in a table (or a figure) such as the green dot should be explained in the explaining text or legend, in this case: green dot, calculated mean.

Reply: Done

Captions for all figures were updated

Congenital malformation, I accept your explanation. Line 21 change “form” to “from”.

Reply: Done

Line 28 change “form” to “from”.

Reply: Done

Maternal hypoglycemia. All three studies in the meta-analysis are assessed by the authors as having low risk of bias which means that they are of high quality although they have low numbers of events. This could increase the risk of bias. More importantly, the oldest (Steel et al) was published 1990 with data from the nineteen seventies and eighties before the meticulous surveillance of b-glucose and new recombinant, ultra-rapid acting insulins were introduced in contrast to Holmes et al. published 2017 and Temple et al who used data from 1990 to 2002. The heterogeneity could thus be caused by secular trends in treatment and surveillance. The 2 modern studies (Holmes and Temple) show no significant difference between PCC and controls. This implicates that maternal hypoglycemia in modern management of type 1 diabetic women (at least in high income countries) is not influenced by PCC. If the authors still consider Holmes and Temple having low risk of bias, I think that the evidence for this outcome should be low-quality, not very-low quality. I would say that you are not totally uncertain if PPC confers hypoglycemia or not compared to non-PCC, but it might not matter, still more studies could alter the evidence.

Reply: Done, we agree with the reviewer that more studies are needed to improve certainty

I think your explanation of the result concerning HbA1c applies here. The explanation of few studies shall be omitted from the text. I presume that your statement of true effect means that using modern data and more research would show if the risk of hypoglycemia is increased or decreased using PCC in a time with continuous glucose measuring (CGM) for an increasing part of patients with type 1 diabetes. You have already mentioned this in implications for research.

Reply: agree

Preterm delivery. Omit the sentence “The grade of evidence is considered moderate due to the small effect”. A small effect could be the true outcome. Adjust sentence in lines 52-53 …selective reporting increase our confidence in the outcome of a small reduction (4%) in preterm delivery.

Reply: Done

The sentence was omitted and the second sentence was adjusted.

Perinatal mortality. The upgrading is a too high. A reduction of RR to < 0.5 as in the current meta-analysis increase the grade by one star to moderate quality of evidence. RR < 0.2, which is not the case here, gives rise to two stars. The quality of evidence should thus be moderate.

Reply: Corrected in abstract, results and discussion

Small for gestational age. The RR is 0.52 not reaching the level <0.50 for upgrading one star. Thus up-grading is not an obvious action. If most included studies have low risk of bias the default grade score could be moderate quality. Upgrading to high is not appropriate here.

Reply: Corrected in abstract, results and discussion.

Other outcomes. Many of these outcomes are based on studies with low risk of bias. As the authors point out in the reply the CI of most studies cross the line depicting no significant difference between the PCC and non-PCC groups, which means that PCC might have no effect on these outcomes. Many studies with low bias and no difference between groups could render an upgrading, but I can accept your statement of low-certainty evidence.

Reply: Agree

Discussion.

The impact of PCC might be less or even non-existent in some outcomes. This should be summarized in the first paragraph as it is valuable knowledge for maternity health care planners. The risk for hypoglycemia should be elaborated further as one of the included studies describes management no longer used in high income countries.

Lines 142- 143. Alter to … “moderate quality albeit the effect was size was small”.

The authors point to a very important issue in studies with a great time span. Practice changes over time (i.e. different criteria for diabetes diagnosis, and introduction of continuous glucose measurements) this is especially important for the risk of hypoglycemia during pregnancy, which was much discussed earlier when stricter management began. I persists in the suggestion that you address this, even if you do no sensitivity analysis as this an important outcome that might need further studies with modern treatment. This should be addressed in connection to the last paragraph on page 40, lines 193-195. See also my comments in the result section.

Reply: Done

This paragraph was added to the discussion section

Meta-analysis results showed that the effect of PCC on maternal hypoglycemia was of low-certainty level. The three included studies had a low risk of bias but a high level of heterogeneity. Studies included were conducted in different time periods, during which tremendous changes were made with the management of diabetes, which explains the heterogeneity level. Evidence could be altered if more studies were to be carried out. 

Sensitivity analysis was added according to level of quality of included studies (supplementary 4)

The result of this review is rather clear. PCC increases the health of mothers with diabetes and perinatal period of the off-spring in high-income countries. The authors describe some studies using surveys and qualitative design to investigate women’s attitudes towards pregnancy planning in Britain and the US and one study of risk factors for SGA in Brazil. The results from these studies might help formulate strategies to reach women with diabetes in childbearing age at risk, in underprivileged areas, and in low-income countries. This could be addressed in the section implications for research.

Reply: Done

This paragraph was added to implication to research

There is lack of studies addressing these problems in the low-income countries that raise the need for future research both quantitative and qualitative.

General remarks.

A further linguistic revision of the text is needed and there are still typos in the text.

Reply: Done

Reviewer #2: With the large amount of data, the authors should check carefully the numbers in text and tables.

Abstract

The effect on maternal hypoglycemia is reported to be RR 1,42;95% CI: 0,72-2,82 (p 2, l 60) and also in the Effects of intervention (p 32, l 36), while in Tables 5 and 6 it is said to be RR 1,38; 95% CI 1,07-1,79.

Reply: Corrected according to the tables

Search methods

..all the literature published up to March 2019 ..

This sentence might have included "between 1983 and .." - this information is given in Study Characteristics (p 10, l 268)., One study dated 1982 is also included, maybe this one was found linked to another study?

Reply: We mean studies from the commencement of each database till March 2019 

Table 1

Cohort study 12 (Garcia-Patterson): a miscarriage rate of 13/66 gives 53 continuing pregnancies. The rate of SGA should probably read 1/53.

Similarly, the rate of RDS should probably read 12/119.

Reply: corrected

Table 2

Macrosomia rate in the NO-PCC group should probably read 4/12.

Reply: corrected 

Assessment of the methodological quality of the included studies

Ref (40) is named Rosen et al (p 26, l 327). The correct spelling is Rosenn, as given in References, Tables 1 and 4.

Reply Corrected

Effects of intervention

Here the same reference is named Rossen (p 31, l 9).

Reply: Corrected

Gestational age at booking for antenatal care

MD 1,31(probably weeks? as it is translated into approximately ten days) - p 31, l 7.

Reply: Yes, it was computed as weeks in the mata-analysis 

Discussion

Incremental increase ... (p 38, l 134) is superfluous.

Reply: Corrected to this sentence

“Many studies confirmed the incremental increase in the rate of adverse pregnancy outcomes with the increase in the level of HbA1c”

FORM should be corrected to FROM at several places (p 31, l 21, p 32, l 28 and 51, p 33, l 73)

Reply: Corrected

Reviewer #3: Conduct a meta-analysis to evaluate the effectiveness and safety of pre-conception care in improving maternal and perinatal outcomes and evaluate the grade of the body of evidence for each outcome. They identify 36 studies and the meta-analysis results showed that PCC results in large reduction in congenital malformations, lowers HaA1c in the first trimester of pregnancy, lowers the preterm delivery rate

1. Abstract: “the result… that PCC results in large reduction in congenital malformations, …owers HbA1c….” the causal effect was implied in the results. However, the study includes both trial and observational studies. The causal effect should be avoided throughout the manuscript.

Reply: revised.

2. Line 311. 21 studies were assigned to be at low risk of bias while 15 studies at high risk of bias. A sensitive analysis may be warranted to evaluate the robustness of the findings.

Reply: Done

• Sensitivity analysis according to the quality of included studies was done, all forest plots are added to Supplementary file 4, method of sensitivity analysis was added to the methods section and results of sensitivity analysis was added to the results section.

3. Figures 2 and 9. What’s the green rectangle in the figure 2 or blue rectangle in figure 9? Explanation to the figures are needed 

Reply: explanation was added to all figures.

---

## [Decision Letter · Decision Letter 2]

20 Jul 2020

PONE-D-19-30647R2

Systematic review and meta-analysis of the effectiveness of pre-pregnancy care for women with diabetes for improving maternal and Perinatal outcomes

PLOS ONE

Dear Dr. Fayed,

We consider that this revised version is considerably improved, although some minor points need to be addressed before it can be accepted, according to reviewer's 1 last remarks (below).   

Please submit your reply and final revision of the manuscript by Sep 03 2020 11:59PM. If you will need more time than this to complete your revisions, please reply to this message or contact the journal office at plosone@plos.org. Please include the following items when submitting your revised manuscript:

We look forward to receiving your revised manuscript.

Kind regards,

Umberto Simeoni

Academic Editor

PLOS ONE

Reviewers' comments:

Reviewer's Responses to Questions

**Comments to the Author**

1. If the authors have adequately addressed your comments raised in a previous round of review and you feel that this manuscript is now acceptable for publication, you may indicate that here to bypass the “Comments to the Author” section, enter your conflict of interest statement in the “Confidential to Editor” section, and submit your "Accept" recommendation.

Reviewer #1: (No Response)

Reviewer #3: All comments have been addressed

2. Is the manuscript technically sound, and do the data support the conclusions?

Reviewer #1: Partly

Reviewer #3: (No Response)

3. Has the statistical analysis been performed appropriately and rigorously? 

Reviewer #1: Yes

Reviewer #3: (No Response)

4. Have the authors made all data underlying the findings in their manuscript fully available?

Reviewer #1: Yes

Reviewer #3: (No Response)

5. Is the manuscript presented in an intelligible fashion and written in standard English?

Reviewer #1: Yes

Reviewer #3: (No Response)

6. Review Comments to the Author

Reviewer #1: Comments to authors.

Dr Amel Fayed (corresponding author) and co-workers have answered to my review and done a lot of more changes that are satisfying and thus increased the clarity of the manuscript. Still there are some issues that need to be addressed.

Title. Perinatal should be in lower case “perinatal”.

Methods.

The caption of the GRADE table.

e. I don’t agree that the number of studies is low. Three studies can be good enough especially if they have low bias. The problem here is that the study with the greatest difference between intervention and control is very old and reflect a surveillance of blood glucose no longer in use. I suggest a wording such as “Downgraded to very low-level of certainty as there was a high heterogeneity due to major changes in surveillance of diabetes between the studies”.

Results.

I can´t see that the green dots in the figures are explained. Add an explanation in the captions of all illustrations.

Maternal hypoglycemia.

In table 6, Summary of findings, you state that PCC has no effect on hypoglycemia but in fact the RR is 1.38 (1.07-1.79) a significant difference. So the conclusion is that it seems to have an effect. You explain this finding in the Discussion in a satisfying way. In the text of the result section (page 33, line 36) I suggest that you alter the text to “PCC seems to have an effect on hypoglycemia during the first….) and omit small number of studies and emphasize the great span of time between studies one of which used blood glucose surveillance no longer in use.

Discussion.

I can´t find the sensitivity analysis of maternal hypoglycemia that according to the authors is included in supplement 4.

There are still some typos in the manuscript. For instance an extra dot in row 167, page 41. In table 4 “Risk of bias assessment of the included studies (Steel 1982,1990) right column what is meant? The low risk due to age difference and number of smokers between the groups no regression??

In the text under Table 4 (page 31) Studies at “high risk of bias” score less than six stars or score no stars ….”

You need to scrutinize the manuscript for errors.

Reviewer #3: (No Response)

7. PLOS authors have the option to publish the peer review history of their article (what does this mean?). If published, this will include your full peer review and any attached files.

Reviewer #1: No

Reviewer #3: No

---

## [Author Response · Author response to Decision Letter 2]

22 Jul 2020

Dear Editor,

We would like to thank our reviewers for their constructive comments, and here are our point-by-point replies:

Title. Perinatal should be in lower case “perinatal”.

Reply: Done

Methods.

The caption of the GRADE table.

e. I don’t agree that the number of studies is low. Three studies can be good enough especially if they have low bias. The problem here is that the study with the greatest difference between intervention and control is very old and reflect a surveillance of blood glucose no longer in use. I suggest a wording such as “Downgraded to very low-level of certainty as there was a high heterogeneity due to major changes in surveillance of diabetes between the studies”.

Reply: Corrected to high heterogeneity and inconsistency of direction of effect as the point estimate of each study is indifferent place in the forest plot 

Results.

I can´t see that the green dots in the figures are explained. Add an explanation in the captions of all illustrations.

Reply: All graphs have explanation as the following

“The large green square represents the estimate effect of the study with the highest weight and very precise 95% CI. The black diamond represents the pooled difference estimate. Heterogeneity is quantified by I2 statistics, an I2 value ≥ 50 indicates substantial heterogeneity. Estimated results are presented as mean difference with 95% Confidence Interval. PCC= Preconception care; No PCC= No preconception care; CI= Confidence intervals.”

Maternal hypoglycemia.

In table 6, Summary of findings, you state that PCC has no effect on hypoglycemia but in fact the RR is 1.38 (1.07-1.79) a significant difference. So the conclusion is that it seems to have an effect. You explain this finding in the Discussion in a satisfying way. In the text of the result section (page 33, line 36) I suggest that you alter the text to “PCC seems to have an effect on hypoglycemia during the first….) and omit small number of studies and emphasize the great span of time between studies one of which used blood glucose surveillance no longer in use.

Reply: The paragraph was changed into the following paragraph

“We are uncertain about the effect of PCC on maternal hypoglycemia during the first trimester of pregnancy; (RR 1.38; 95% CI: 1.07- 1.79); three studies; 686 women; very low-certainty evidence) (Fig 5) (table 5). The grade of evidence was downgraded from low to very low due to inconsistency of the direction of effect and high heterogeneity (I2 =76%) in the included studies (table 6). The true effect is likely to be substantially different from the effect estimated in this review.”

Discussion.

I can´t find the sensitivity analysis of maternal hypoglycemia that according to the authors is included in supplement 4.

Reply : The three studies included in the assessment of hypoglycemia were all of low-risk of bias and the heterogeneity is explained by the long time span of the included studies, that is why we did not conduct sensitivity analysis for the maternal hypoglycemia outcome because we restricted the sensitivity analysis to outcomes with studies with high risk of bias as you have suggested. However, we decided to exclude the oldest study from the meta-analysis to support our discussion of its effect and not as sensitivity analysis, please see the Forest Plot below (S4 figure 8)

Supplementary Fig 8. Risk ratio for maternal hypoglycemia from three studies of women with pre-existing diabetes mellitus who did or did not receive preconception care.

Data of Steel 1990 were not estimated in the analysis. The large blue square represents the estimate effect of the study with he highest weight and very precise 95% CI. The black diamond represents the pooled risk estimate. Heterogeneity is quantified by I2 statistics, an I2 value ≥ 50 indicates substantial heterogeneity. Estimated results are presented as risk ratio with 95% Confidence Interval. PCC= Preconception care; No PCC= No preconception care; CI= Confidence intervals.

There are still some typos in the manuscript. For instance, an extra dot in row 167, page 41. 

Reply: corrected

In table 4 “Risk of bias assessment of the included studies (Steel 1982,1990) right column what is meant? The low risk due to age difference and number of smokers between the groups no regression??

Reply: Corrected 

There is no clinical difference between the studied groups regarding the age (27 vs 25), there is difference in other confounders such as number of smokers and no adjustment or regression analysis was done, so this study lost one stars in comparability and considered at “Low risk of bias”.

In the text under Table 4 (page 31) Studies at “high risk of bias” score less than six stars or score no stars ….”

Reply: Corrected to 

Studies at “high risk of bias” scored less than six stars or scored no stars in the comparability domain, irrespective of the number of stars scored.

---

## [Editor Report · Decision Letter 3]

30 Jul 2020

Systematic review and meta-analysis of the effectiveness of pre-pregnancy care for women with diabetes for improving maternal and perinatal outcomes

PONE-D-19-30647R3

Dear Dr. Fayed,

We’re pleased to inform you that your manuscript has been judged scientifically suitable for publication and will be formally accepted for publication once it meets all outstanding technical requirements.

Kind regards,

Umberto Simeoni

Academic Editor

PLOS ONE
---

## [Editor Report · Acceptance letter]

3 Aug 2020

PONE-D-19-30647R3 

Systematic review and meta-analysis of the effectiveness of pre-pregnancy care for women with diabetes for improving maternal and perinatal outcomes 

Dear Dr. Fayed:

I'm pleased to inform you that your manuscript has been deemed suitable for publication in PLOS ONE. Congratulations! Your manuscript is now with our production department. 

Kind regards, 

on behalf of

Dr. Umberto Simeoni 

Academic Editor

PLOS ONE